# SNHG5 promotes colorectal cancer cell survival by counteracting STAU1-mediated mRNA destabilization

Nkerorema Djodji Damas[1], Michela Marcatti[1], Christophe Côme[1], Lise Lotte Christensen[2], Morten Muhlig Nielsen[2], Roland Baumgartner[1], Helene Maria Gylling[1], Giulia Maglieri[1], Carsten Friis Rundsten[1], Stefan E. Seemann[3], Nicolas Rapin[1], Simon Thézenas[4], Søren Vang[2], Torben Ørntoft[2], Claus Lindbjerg Andersen[2], Jakob Skou Pedersen[2] & Anders H. Lund[1]

We currently have limited knowledge of the involvement of long non-coding RNAs (lncRNAs) in normal cellular processes and pathologies. Here, we identify and characterize SNHG5 as a stable cytoplasmic lncRNA with up-regulated expression in colorectal cancer. Depletion of SNHG5 induces cell cycle arrest and apoptosis in vitro and limits tumour outgrowth in vivo, whereas SNHG5 overexpression counteracts oxaliplatin-induced apoptosis. Using an unbiased approach, we identify 121 transcript sites interacting with SNHG5 in the cytoplasm. Importantly, knockdown of key SNHG5 target transcripts, including SPATS2, induces apoptosis and thus mimics the effect seen following SNHG5 depletion. Mechanistically, we suggest that SNHG5 stabilizes the target transcripts by blocking their degradation by STAU1. Accordingly, depletion of STAU1 rescues the apoptosis induced after SNHG5 knockdown. Hence, we characterize SNHG5 as a lncRNA promoting tumour cell survival in colorectal cancer and delineate a novel mechanism in which a cytoplasmic lncRNA functions through blocking the action of STAU1.

[1] Biotech Research and Innovation Centre, University of Copenhagen, Ole Maaloes Vej 5 2200 Copenhagen, Denmark. [2] Department for Molecular Medicine, Aarhus University Hospital, Brendstrupgaardsvej 21 8000 Aarhus, Denmark. [3] Center for Non-Coding RNA in Technology and Health, University of Copenhagen, Grønnegårdsvej 3 1870, Frederiksberg, Denmark. [4] Biostatistics Unit, Institut Régional du Cancer de Montpellier (ICM)—Val d'Aurelle, 208 Avenue des Apothicaires, 34298 Montpellier, France. Correspondence and requests for materials should be addressed to A.H.L. (email anders.lund@bric.ku.dk).

Aberrant proliferation and increased cell survival are key processes during malignant transformation and tumour progression. To facilitate these processes, a number of chromosomal rearrangements, mutations and epigenetic modifications are typically selected for in cancerous cells, resulting in an overall alteration of gene expression. Colorectal carcinoma (CRC) is the third most common cancer worldwide[1]. While invasive colorectal cancers that have not yet compromised regional lymph nodes (stage I–II) have relatively good prognoses with the current treatments and are curable in 73% of the cases, the progression of the disease is fast and untreated tumours rapidly disseminate to lymph nodes (stage III) and metastasize to distant sites (stage IV)[2]. Thus, a better understanding of the mechanisms driving the disease and identification of additional therapeutic targets is a priority for improving CRC treatment[3]. Several studies have pointed to the emerging roles of long non-coding RNAs (lncRNAs) in tumour development, which could provide new candidates for diagnostics and therapy.

Although mammalian genomes are widely transcribed, only 1–2% of the genomic output encodes for proteins[4,5]. Among the large fraction of non-coding transcripts, the class of lncRNAs, arbitrarily defined as transcripts longer than 200 nts, is receiving increasing attention and may present new opportunities for cancer diagnosis and treatment. Although thousands of lncRNAs have been identified, we still lack insight into the structural–functional significance of the vast majority of these molecules in regulating fundamental cellular processes. However, extensive gene expression and copy number variation analyses have linked alteration of lncRNA expression to tumour development. Resultantly, several lncRNAs, such as MALAT1, HOTAIR, ANRIL, PVT1 and lincRNA-p21, have been reported to play significant roles in cancer initiation and development[6–10].

In this study, we aimed to identify and characterize lncRNAs functionally impacting on CRC. Profiling a large set of CRCs, we identified the cytoplasmic lncRNA SNHG5 as significantly overexpressed in tumours. We provide evidence that SNHG5 expression regulates the survival of CRC cells and the progression of CRC tumour xenografts in a mouse model. Importantly, whereas knockdown of SNHG5 prominently induces apoptosis, SNHG5 overexpression can protect CRC cells from oxaliplatin-induced apoptosis. While most of the hitherto characterized lncRNAs function in the nucleus, much less is known about lncRNAs and their mode of action in the cytoplasmic compartment. A notable exception being competing endogenous RNAs (ceRNAs), which act as molecular sponges for microRNAs hence relieving repression of target mRNAs[11,12]. Other known mechanisms for lncRNAs in the cytoplasm involve post-transcriptional regulation affecting mRNA stability or accessibility to the translational machinery[13]. Through an unbiased forward identification of mRNAs interacting with SNHG5 in the cytoplasm, we identify 121 interacting transcript sites in HCT116 CRC cells. Importantly, loss of SNHG5 reduces the protein levels of the interactors via destabilization of their mRNAs. We further characterize the interaction of SNHG5 with the target SPATS2, and demonstrate that loss of SPATS2 phenocopies the effect of SNHG5 depletion. STAU1 is a part of a highly conserved family of double-stranded RNA-binding proteins implicated in mRNA transport, stability and translation[14–17]. We show here that SNHG5 binding to target mRNAs protects these from STAU1-mediated degradation. Importantly, loss of STAU1 also rescues the apoptotic effect of SNHG5 depletion.

Hence, we here provide new insights into the significance of lncRNAs in CRC in general and to the specific role of SNHG5 in promoting CRC cell survival.

## Results

**SNHG5 is up-regulated in colorectal cancer.** To identify non-coding RNAs deregulated in CRC, we profiled their expression in a cohort of 44 carcinomas, 39 adenomas, 20 adjacent normal mucosa and 10 CRC cell lines using a previously described custom-designed microarray platform[18]. As expected, the overall expression level of lncRNAs was lower than that of the protein-coding genes in all sample sets (Supplementary Fig. 1a)[19].

Among the most significantly deregulated transcripts, we identified several lncRNAs previously found deregulated in cancer, such as ncRAN and GAS5, indicating the validity of our approach (Supplementary Fig. 1b,c). We focus here on SNHG5 for which no function in CRC has been ascribed so far. RNA-seq confirmed the up-regulation of SNHG5 in CRC in a larger independent cohort of 294 normal colon mucosa samples, 33 adenomas and 280 CRC samples ($n = 313$) (Fig. 1a). This was confirmed also restricting the analysis to the 268 paired samples included in the same cohort (Supplementary Fig. 2a). Interestingly, we find SNHG5 to be significantly up-regulated both between normal tissues and adenomas, and from adenomas to carcinoma stage I (malignant tumour), suggesting SNHG5 up-regulation as an early event in CRC development (Fig. 1b).

SNHG5 is composed of six exons spliced into two main isoforms: the 1032 nts long SNHG5-1 and the shorter SNHG5-2 (430 nts) (Fig. 1c). SNHG5-2 is the most abundant isoform expressed in CRC (Supplementary Fig. 2b), and we refer to it as SNHG5 in the following experiments. Two snoRNAs, U50A and U50B, are encoded in introns 4 and 5, respectively. We profiled the expression of SNHG5 in a panel of human tissues and found it most highly expressed in muscles, liver and heart (Supplementary Fig. 2c). Reverse transcription (RT)–PCR analysis revealed that the SNHG5 transcript, like many other mRNA-like non-coding RNAs, is poly-adenylated (Supplementary Fig. 2d). The SNHG5 transcript has a half-life ($t½$) of 1 h, indicating an intermediate turnover rate when compared with transcripts with a very short half-life, such as MYC ($t½ = 30$ min), or more stable mRNAs, such as ACTB ($t½ = 5–6$ h) (Supplementary Fig. 2e). RNA fluorescence in situ hybridization (RNA-FISH) was performed using LNA probes spanning the exon–exon junctions 3–4 and 4–5, respectively, and showed SNHG5 to be predominantly cytosolic in both HCT116 CRC cells and in immortalized TIG-3 human fibroblasts (Fig. 1d). This was also demonstrated by subcellular fractionation studies and qRT–PCR (Fig. 1e). Polysome fractionation studies showed a clear SNHG5 enrichment in the light, untranslated fraction suggesting no association with ribosomes (Supplementary Fig. 2f). In conclusion, we characterize SNHG5 to be a stable and poly-adenylated ncRNA transcript not associated with ribosomes and with an up-regulated expression in CRC.

**SNHG5 affects proliferation and survival of CRC cells.** Towards identifying a function for SNHG5 we profiled its expression in a panel of CRC cell lines (Supplementary Fig. 3a) and designed two independent short interfering RNAs (siRNAs) targeting SNHG5 with knockdown efficiencies >75% (Fig. 1f). The cytosolic fraction of the transcript was the most strongly reduced when analysed 36 h after siRNA transfection (Supplementary Fig. 3b). Moreover, the expression levels of the U50A snoRNA remained unchanged, suggesting that the siRNAs target the spliced transcript and that phenotypic effects will be independent of the snoRNAs, which in these cell lines exerts a nuclear restricted expression (Supplementary Fig. 3c,d). To investigate biological processes affected by SNHG5 depletion, RNA-seq was performed in HCT116 cells 36 h after transfection with two independent siRNAs against SNHG5 or a negative control siRNA

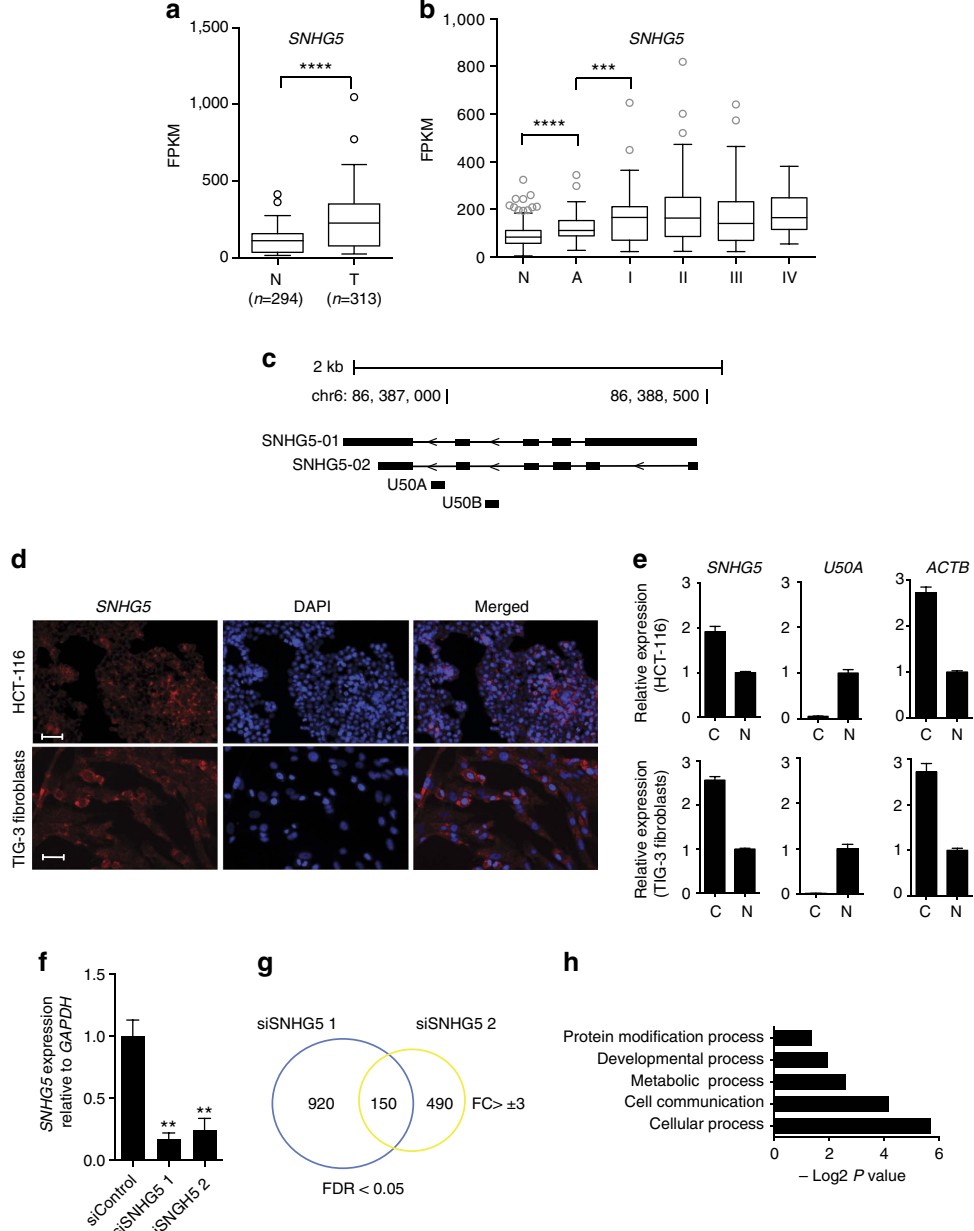

**Figure 1 | SNHG5 is deregulated in CRC.** (**a**) The expression levels of *SNHG5* were profiled in clinical specimens by RNA-seq, where the RNA was extracted from tumour (C) or adjacent normal tissues (N) (****$P$ value $< 0.0001$, Mann–Whitney $t$-test). (**b**) Box plot representing stratified *SNHG5* expression in CRC clinical specimens (N = normal adjacent tissue, A = adenomas, I–IV = CRC progression stages). The bottom line of each box represents the 1st quartile of sample population, the top line of the box represents the third quartile, while the middle quartile represents the population median. (***$P$ value $< 0.001$, ****$P$ value $< 0.0001$, Mann–Whitney $t$-test). (**c**) Schematic representation of the *SNHG5* locus. (**d**) RNA-FISH of *SNHG5* in red (Cy3) with a nuclear DAPI stain (blue) in HCT116 and TIG-3 cells. Scale bar, 50 μm. (**e**) RNA from the nuclear (N) and cytosolic (C) fractions were isolated from HCT116 and TIG-3 cells and *SNHG5* levels measured using qRT–PCR. *U50A* and *ACTB* were used as nuclear and cytosolic controls, respectively. The expression values for each transcript are relative the total RNA input and normalized on the nuclear fraction expression levels. Error bars represent s.e. from one representative experiment. (**f**) qRT–PCR. HCT116 cells were transfected with 50 nM siSNHG5-1-2 or control siRNA and the RNA collected after 36 h. *SNHG5* expression data are showed normalized to the *GAPDH* housekeeping gene and normalized to the non-targeting control siRNA. (**g**) Venn diagram representing the number of genes altered in HCT116 cells following siSNHG5-1-2 transfection after 36 h (log2 fold change $> \pm 3$, FDR $< 0.05$). (**h**) Gene ontology annotation. Statistical overrepresentation test (Bonferroni-corrected) was applied to identify gene categories affected by *SNHG5* depletion.

(Fig. 1g). We found 150 genes to be significantly deregulated by both *SNHG5* siRNAs (fold change $>3$, FDR $<0.05$). Gene ontology analysis revealed a significant enrichment of genes involved in differentiation processes, such as mesoderm development, cell communication and metabolic processes (Fig. 1h). Furthermore, gene set enrichment analysis[20] suggested that

*SNHG5* depletion impacts key survival pathways in CRC cells, such as the STAT3 pathway (Supplementary Fig. 3e). This was further confirmed using western blots for STAT3, a known STAT3 target gene BCL-XL, and BCL-2 (Supplementary Fig. 3f). Importantly, knockdown of *SNHG5* (Supplementary Fig. 3g) resulted in a strong anti-proliferative effect in CRC cells (Fig. 2a).

This result was confirmed by cell cycle analysis using EdU incorporation, which revealed a significant decrease in the S-phase cell population in *SNHG5*-depleted cells (Fig. 2b). To investigate an impact of *SNHG5* on survival, CRC cell lines were stained for cleaved Caspase-3 protein and analysed by flow cytometry, which revealed a strong induction of apoptosis in both HCT116, CACO-2 and DLD-1 cells (Fig. 2c, top). This was confirmed by western blots for cleaved PARP-1 (Fig. 2c, bottom).

As siRNA depletion studies may suffer from off-target effects, we subsequently investigated the effect of *SNHG5* overexpression. Towards this, both a cDNA version (430 bp) and the entire genomic locus of *SNHG5* (1800 bp) were cloned (Fig. 2d) and transiently expressed in HT-29 cells, which display a lower endogenous *SNHG5* level as compared with HCT116, CACO-2 and DLD-1 cells (Supplementary Fig. 3a). Forty-eight hour after transfection a 60-fold average increase in the *SNHG5*

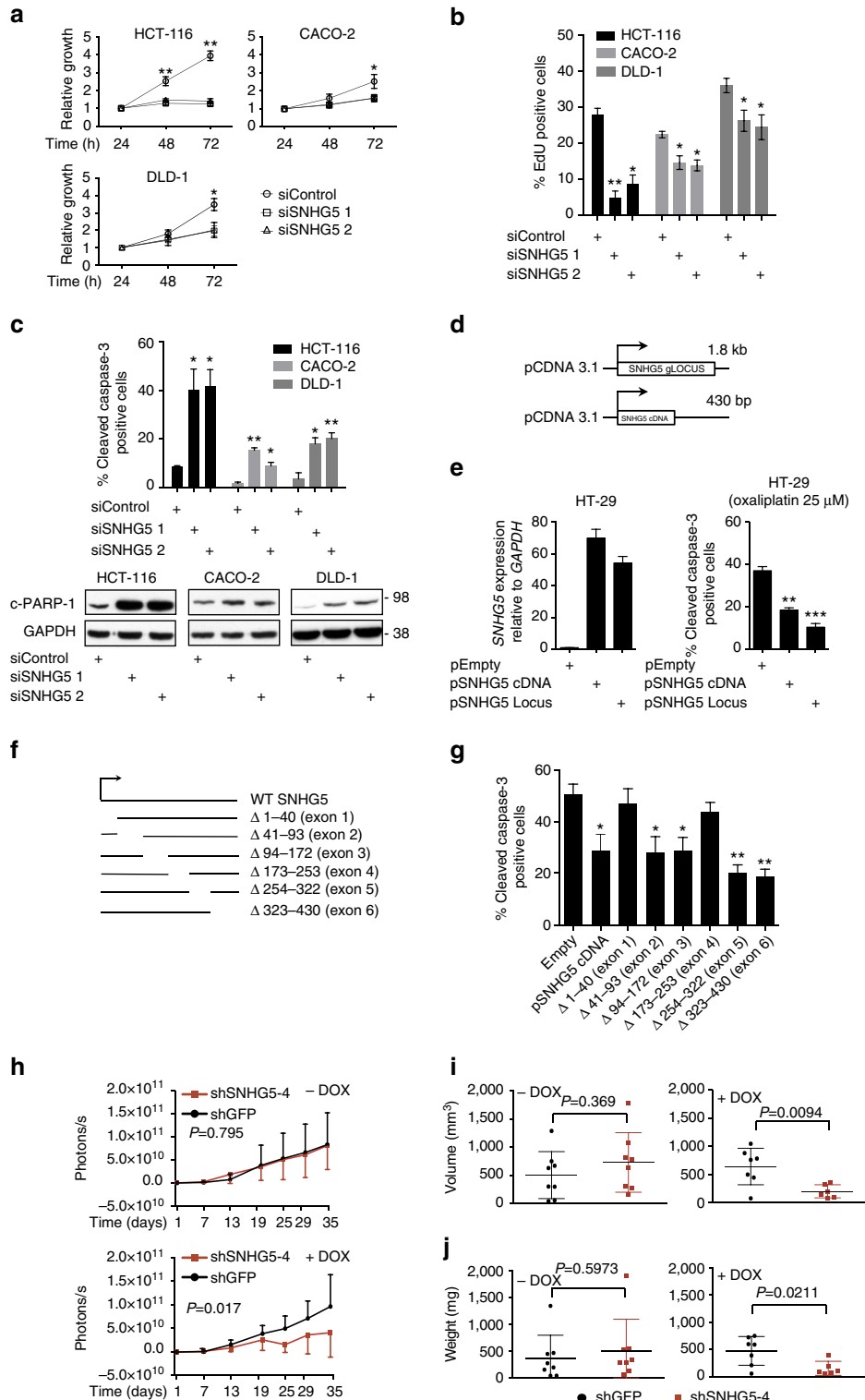

levels were detected by qRT–PCR in comparison to the empty vector control (Fig. 2e, left). The cells were subsequently treated for 24 h with oxaliplatin, a drug commonly employed in CRC treatment. Interestingly, using flow cytometry, we observed a significant decrease in the cleaved Caspase-3 positive population of HT-29 cells overexpressing *SNHG5* (Fig. 2e right). This protective effect was evident for both the cDNA construct and the entire *SNHG5* locus, suggesting no significance of the snoRNAs in this assay. No proliferative effect was observed in HT-29 cells overexpressing the *SNHG5* lncRNA in the absence of drug treatment (Supplementary Fig. 4a,b). Next, we created a panel of *SNHG5* mutants lacking individual exons and tested the ability of these mutants to protect HT-29 cells from oxaliplatin-induced apoptosis (Fig. 2f). Interestingly, whereas mutants for exons 2, 3, 5 and 6 all protected the cells to the same extend as full-length *SNHG5*, mutants lacking either exon 1 or exon 4 failed to protect against oxaliplatin-induced apoptosis (Fig. 2g).

We next employed a luminescent xenograft mouse model to query whether *SNHG5* expression is required for tumour growth *in vivo* (Supplementary Fig. 4c). Luminescent HCT116 CRC cells holding doxycycline-inducible shRNAs targeting *SNHG5* or GFP as control were subcutaneously injected into two groups of immune-compromised mice. To allow time for tumour establishment, doxycycline was added to the drinking water at day 13 after injection. *SNHG5* levels were significantly reduced in the tumours following doxycycline treatment compared with the non-targeting control (Supplementary Fig. 4d). Importantly, we observed a significant negative effect on tumour growth following *SNHG5* depletion (Fig. 2h; Supplementary Fig. 4e). *SNHG5* depletion resulted in a significant reduction of tumour volume and weight as measured at the end of the experiments at day 35 (Fig. 2i,j). Altogether, this data suggest that *SNHG5* expression functions to regulate proliferation and survival of CRC cells both *in vitro* and *in vivo*.

**RIA-seq identifies primary downstream effectors of *SNHG5*.** Until now, only few mechanisms for cytoplasmic lncRNAs have been revealed. To unveil the mechanism by which *SNHG5* affects proliferation and apoptosis in the cytoplasm, we used an unbiased method, RIA-seq (RNA interactome analysis and sequencing), to map transcriptome-wide RNA–RNA interactions[21]. Two sets (Even and Odd) of eleven 3′-biotinylated DNA oligonucleotides were designed to specifically hybridize to *SNHG5*. A luciferase-

specific oligonucleotide pool was used as negative control (Fig. 3a; Supplementary Fig. 5a). The pull-down efficiency was estimated by qRT–PCR to ∼50% of the endogenous *SNHG5* transcript pool in HCT116 cells (Fig. 3b). The interacting RNA transcripts were identified using single-end 50 nucleotide Illumina sequencing and the sequence reads were mapped onto the Hg19 genome scaffold using the STAR aligner tool (see 'Methods' section for details). The relative enrichment of peaks present in both the Even and the Odd pools were calculated with respect to the luciferase pool, which was used as background reference. We identified 121 significant peaks overlapping between the Even and Odd pools ($P$ value < 0.05) (Fig. 3c; Supplementary Fig. 5b). About half of the peaks mapped to coding regions (51.4%) while the majority of the remaining peaks resided in 3′ UTR regions (40%) (Supplementary Fig. 5c).

We ranked a short list of *SNHG5*-interacting mRNAs based on the number of peaks (overlapping and not-overlapping) in each transcript and their local enrichment relative to the luciferase background pool (Supplementary Fig. 5d). The mRNA transcripts representing the peaks were subsequently validated in HCT116 by qRT–PCR using peak-specific primers (Fig. 3d; Supplementary Fig. 5d,e). We identified the *SPATS2*, *PITRM1* and *GLE1* mRNAs as the most enriched *SNHG5*-interacting transcripts. We further validated the RIA-seq data in CACO-2 and DLD-1 cells, where *SNHG5* depletion was also found to affect cell survival (Fig. 2a). In these cells only the interaction with the *SPATS2* mRNA could be consistently confirmed, probably due to lower expression levels of the other target transcripts in these cell lines (Supplementary Fig. 6a,b). Importantly, *SNHG5* depletion results in a clear decrease of the target protein levels (Fig. 3e). To evaluate the effect of *SNHG5* on the stability of the target transcripts, HCT116 cells were treated for 5 h with the RNA PolII inhibitor Triptolide following *SNHG5* knockdown (Fig. 3f). The mRNA stabilities of all three top interactors were reduced on *SNHG5* knockdown as quantified using qRT–PCR. Altogether, this data suggest that *SNHG5* forms RNA–RNA interactions with target mRNAs in the cytoplasm and regulates their stability.

**SPATS2 is a key downstream target of *SNHG5*.** To further explore the underlying mechanism, we focused on the *SNHG5*–*SPATS2* interaction and confirmed that *SNHG5* depletion in

**Figure 2 | Manipulation of *SNHG5* expression regulates proliferation and survival both *in vitro* and *in vivo*.** (**a**) Proliferation assay. CRC cell lines were transfected with 25 nM siSNHG5-1-2 or siControl. Cell numbers were assessed at the designated time points post transfection. Data normalized to the first time point are shown for the mean of 3 independent experiments. Error bars indicate ± s.d. (*$P$ value < 0.05, **$P$ value < 0.01, Student's *t*-test paired). (**b**) HCT116, CACO-2 and DLD-1 cells were transfected as in **a**, pulsed with EdU after 36 h and analysed by flow cytometry to measure the number of EdU-labelled cells representing the S-phase populations. Data represents the mean of three independent experiments. Error bars indicate ± s.e. (*$P$ value < 0.05, **$P$ value < 0.01). (**c**) Top panel: CRC cell lines were transfected as in (**a**). Thirty-six hours after transfection, cells were labelled with cleaved Caspase-3 antibody and analysed by flow cytometry. Data are shown for the mean of three independent experiments. Error bars indicate ± s.e. (*$P$ value < 0.05, **$P$ value < 0.01). Bottom panel: western blot analysis of endogenous cleaved PARP-1 levels in HCT116, CACO-2 and DLD-1 cells transfected with siSNHG5-1-2 or siControl. GAPDH is included as loading control. A representative experiment is shown ($n = 3$). (**d**) Schematic representation of constructs expressing the entire *SNHG5* genomic (pSNHG5 locus) or only the cDNA (pSNHG5 cDNA). (**e**) Left, qRT–PCR. *SNHG5* expression levels were measured 48 h after transfection into HT-29 cells. Data was normalized to the *RPLP0* housekeeping gene and plotted relative to the empty vector control. Error bars indicate the mean ± s.e. of three independent experiments. Right, HT-29 cells were transfected with plasmids as described above. Forty-eight hours after transfection the cells were treated with 25 μg ml⁻¹ of oxaliplatin. Twenty-four hours later the cells were stained with cleaved Caspase-3 antibody and analysed by flow cytometry. Error bars indicate the mean ± s.e. of three independent experiments. (*$P$ value < 0.05, **$P$ value < 0.01, ***$P$ value < 0.01). (**f**) Schematic representation of the *SNHG5* cDNA expressing mutant plasmids. (**g**) Overall, $2 \times 10^5$ HT-29 cells were transfected with mutant plasmids and treated with oxaliplatin as described above. Twenty-four hours later the cells were stained with cleaved Caspase-3 antibody and analysed by flow cytometry. Error bars indicate the mean ± s.e. of three independent experiments. (*$P$ value < 0.05, **$P$ value < 0.01). (**h**) Xenograft growth of HCT116 pFULT-shGFP and HCT116 pFULT-shNHG5-4 cells. Doxycycline was added to the drinking water after 13 days. Mean ± s.d. of tumor luminescence (photons/seconds/cm2) at the indicated time points (for statistic analysis details see 'Methods' section). (**i,j**) Final volume and weight of tumours from the experiment at day 35, respectively.

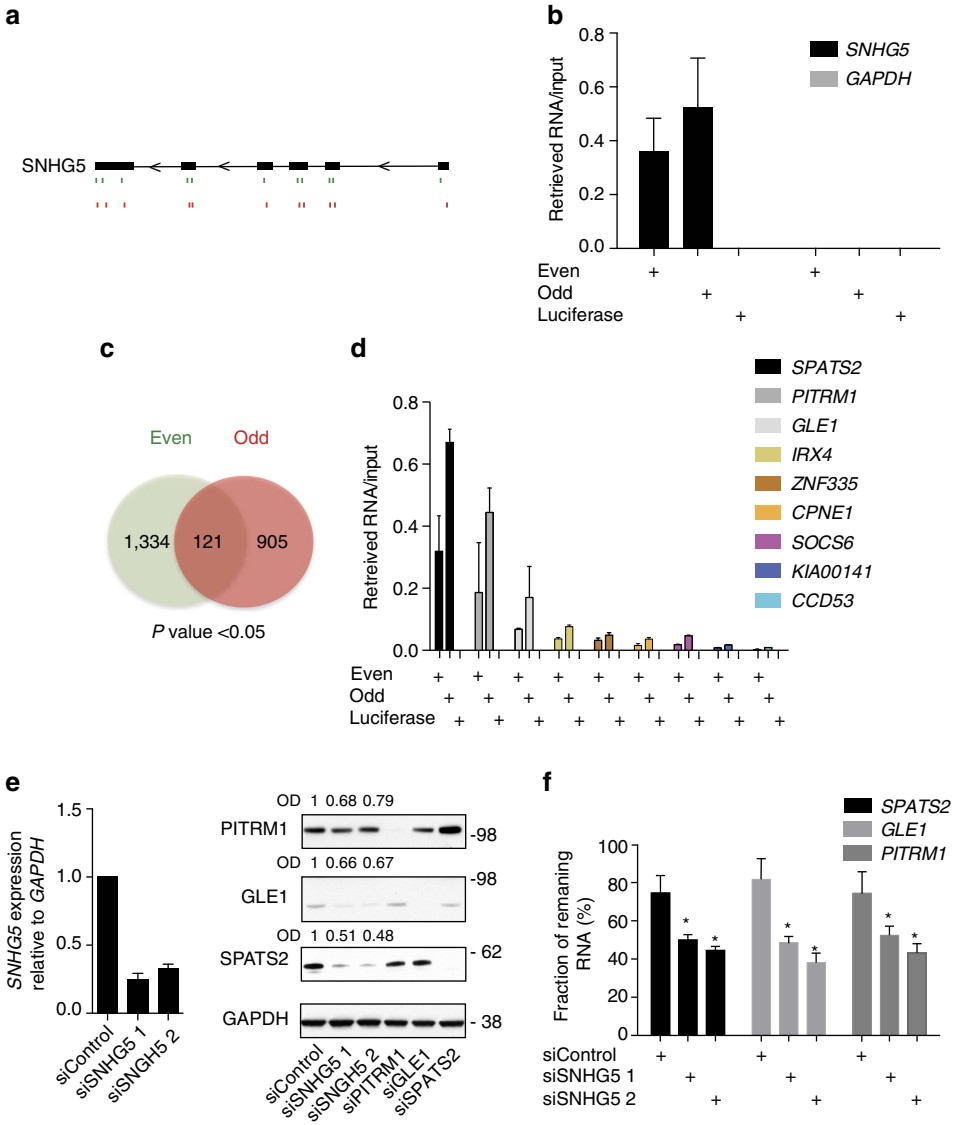

**Figure 3 | SNHG5 targets mRNAs in the cytoplasm. (a)** Schematic representation of biotinylated antisense DNA probes complementary to the *SNHG5* transcript Even (green) and Odd (red) used for the RNA pool-down. **(b)** qRT–PCR was performed to assess the % of *SNHG5* RNA retrieved from HCT116 cell lysates following pull-down with streptavidin beads. Data was normalized to input (1:100) and *GAPDH* mRNA included as negative control. Error bars indicate ± s.d. for the mean of three independent experiments. **(c)** Venn diagram representing the number of *SNHG5* pull-down peaks commonly enriched with both the Even and Odd probe sets. **(d)** Validation of the RIA-seq results by qRT–PCR. Data for each target transcript is normalized on input (1:100). Error bars indicate ± s.d. for the mean of three independent experiments. **(e)** Left, qRT–PCR. *SNHG5* expression levels were measured 48 h after transfection of HCT116 cells. Data was normalized to the *GAPDH* housekeeping gene and plotted relative to the siRNA control. Error bars indicate the mean ± s.e. of three independent experiments. Right, western blot. HCT116 cells 48 h after transfection with the indicated siRNAs. The optical density (OD) of protein bands is indicated relative to corresponding loading control GAPDH and normalized relative the non-targeting siControl. **(f)** Transcript stability of *SNHG5* targets were measured by qRT–PCR in HCT116 cells 36 h after transfection with the indicated siRNAs. The cells were treated with Triptolide at a final concentration of 10 μM for the last 5 h. The mRNA levels are plotted relative to the corresponding expression levels in the untreated cells. Error bars indicate the mean ± s.e. of three independent experiments (Student's *t*-test paired).

HCT116, CACO-2 and DLD-1 cells, where both transcripts are expressed at similar levels (Supplementary Fig. 6c), results in a decrease of the SPATS2 protein levels (Fig. 4a). The SPATS2 gene product is a predicted cytoplasmic RNA-binding protein, but besides its potential involvement in the maturation of gonad cells SPATS2 functions are unknown[22]. Importantly, *SNHG5* over-expression in HT-29 was found to increase SPATS2 protein levels (Fig. 4b). An RNA–RNA interaction site was computatio-nally predicted to span the first exon of *SNHG5* and the 3′ UTR of *SPATS2*. Both the thermodynamics prediction (Energy = − 15.50 kcal mol$^{-1}$; IntaRNA) and the conservation

(PETcofold) point towards these regions as the interaction site. The interaction is only conserved among primates (phylogenetic tree predicted in PETcofold) and the phyloP base-wise conservation (100 vertebrates UCSC genome alignment) shows low evolutionary pressure on *SNHG5* suggesting an evolutionary new binding site (Fig. 4c). As expected, overexpression of *SNHG5* promotes the stabilization of the *SPATS2* mRNA (Supplementary Fig. 6d). To identify the region in *SNHG5* responsible for promoting *SPATS2* mRNA stability, we overexpressed the panel of *SNHG5* mutants and measured the stability of the *SPATS2* mRNA following Triptolite treatment. Interestingly, the *SNHG5*

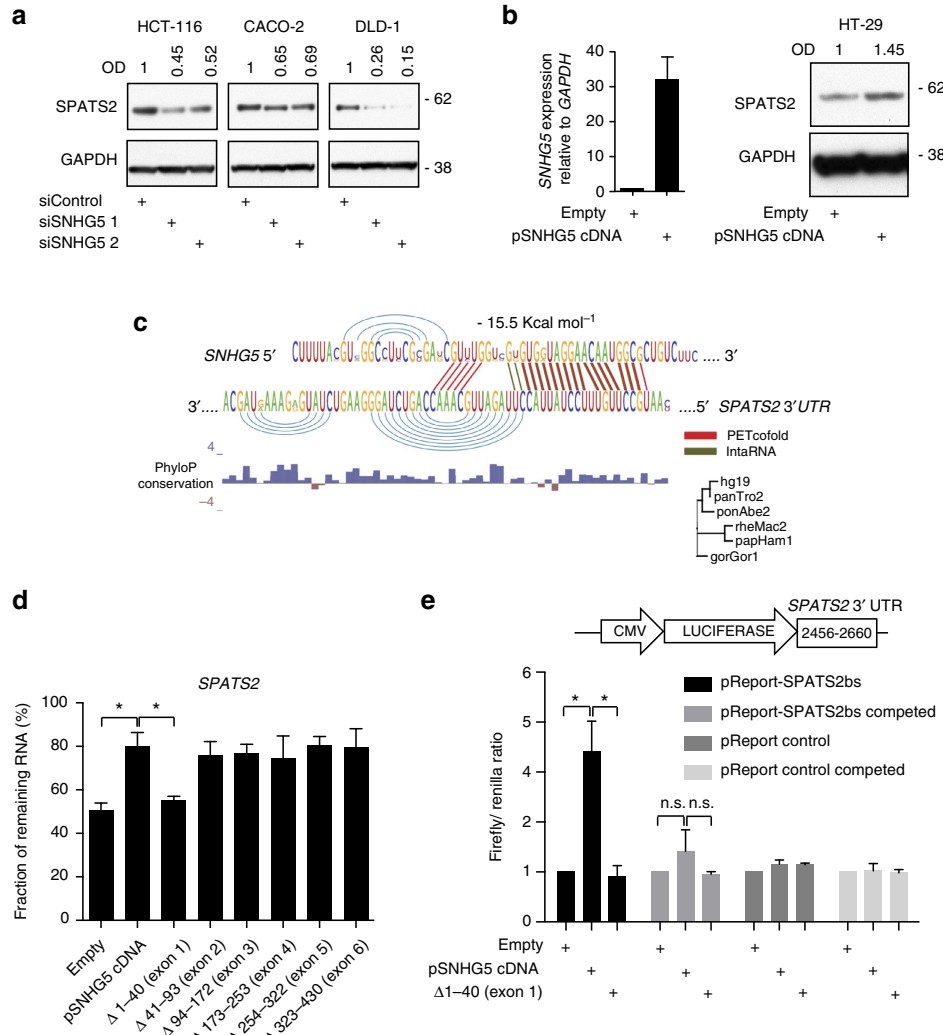

**Figure 4 | SNHG5 promotes SPATS2 mRNA stability.** (**a**) Western blot. HCT116, CACO-2 and DLD-1 cells were transfected with the indicated siRNAs and proteins isolated after 48 h. OD of protein bands is indicated relative to corresponding loading control GAPDH and normalized relative the non-targeting siControl. (**b**) Left, $2 \times 10^5$ HT-29 cells were transiently transfected with the indicated plasmids. *SNHG5* expression levels were measured 48 h after transfection into HT-29 cells using qRT–PCR. Right, western blot. SPATS2 expression levels were evaluated 48 h after transfection. OD of protein bands is indicated relative to corresponding loading control GAPDH and normalized relative the empty vector. (**c**) For graphical output of the interaction between *SNHG5* exon 1 and the *SPATS2* 3′ UTR were combined PETcofold analysis (red) relative to the evolutionary conservation of the interaction site and IntRNA (green) to map the most thermodynamically stable interaction between the lncRNA and the mRNA. Local PhyloP conservation index is included. (**d**) qRT–PCR. $2 \times 10^5$ HT-29 cells were transfected with the indicated plasmids. Forty-eight hours after transfection the cells were treated with Triptolide at a final concentration of 10 μM for 8 h and the RNA subsequently extracted. The percentage retrieved *SPATS2* mRNA was obtained normalizing to the corresponding expression levels in the untreated cells. (**e**) Top, schematic representation of the 3′-UTR region of the *SPATS2* 3′ UTR fragment cloned in the pMIR-REPORT plasmid. Bottom, luciferase reporter assay 48 h after transfection of $2.4 \times 10^4$ HEK293 cells per well (four well each samples) with the indicated plasmids and a renilla luciferase transfection control plasmid. To compete the *SNHG5* binding with the *SPATS2* 3′-UTR, pcDNA 3.1 expressing the complete *SPATS2* cDNA was co-transfected with the indicated plasmids. Error bars indicate ± s.e. of three independent experiments (*$P$ value < 0.05).

mutant lacking exon 1 failed to promote *SPATS2* mRNA stability (Fig. 4d), thus sustaining the notion that *SNHG5* interacts with *SPATS2* via elements in *SNHG5* exon 1.

We then assessed the binding of *SNHG5* to the 3′-UTR of *SPATS2* (nts 2,456–2,660) using a luciferase reporter system (Fig. 4e, top). The ectopic expression of *SNHG5* in HEK293 cells increased the luciferase activity of the *SPATS2* 3′ UTR reporter construct, as compared with the control, whereas the *SNHG5* mutant lacking exon 1 was unable to do so (Fig. 4e, bottom). In addition, we were able to compete the *SNHG5* binding to the *SPATS2* 3′-UTR by co-transfecting the cells with a plasmid vector expressing the full-length *SPATS2* mRNA. In addition, co-transfecting the full-length *SPATS2* mRNA was also sufficient

to compete the *SNHG5* interaction with a second luciferase construct containing the *SNHG5* binding site from the *GLE1* 3′-UTR (nts 2,496–2,706) (Supplementary Fig. 6e,f). Altogether, these data demonstrate that *SNHG5* can directly bind and regulate *SPATS2*.

**The *SNHG5*–*SPATS2* axis promotes CRC cell survival.** SPATS2 has not previously been implicated in cancer-related processes, however, RNA-seq data analysis of CRC-paired samples showed a significant up-regulation of the *SPATS2* transcript in tumours versus correspondent normal tissues (Fig. 5a). Similarly, we find the two other top interactors *GLE1* and *PITRM1* to be

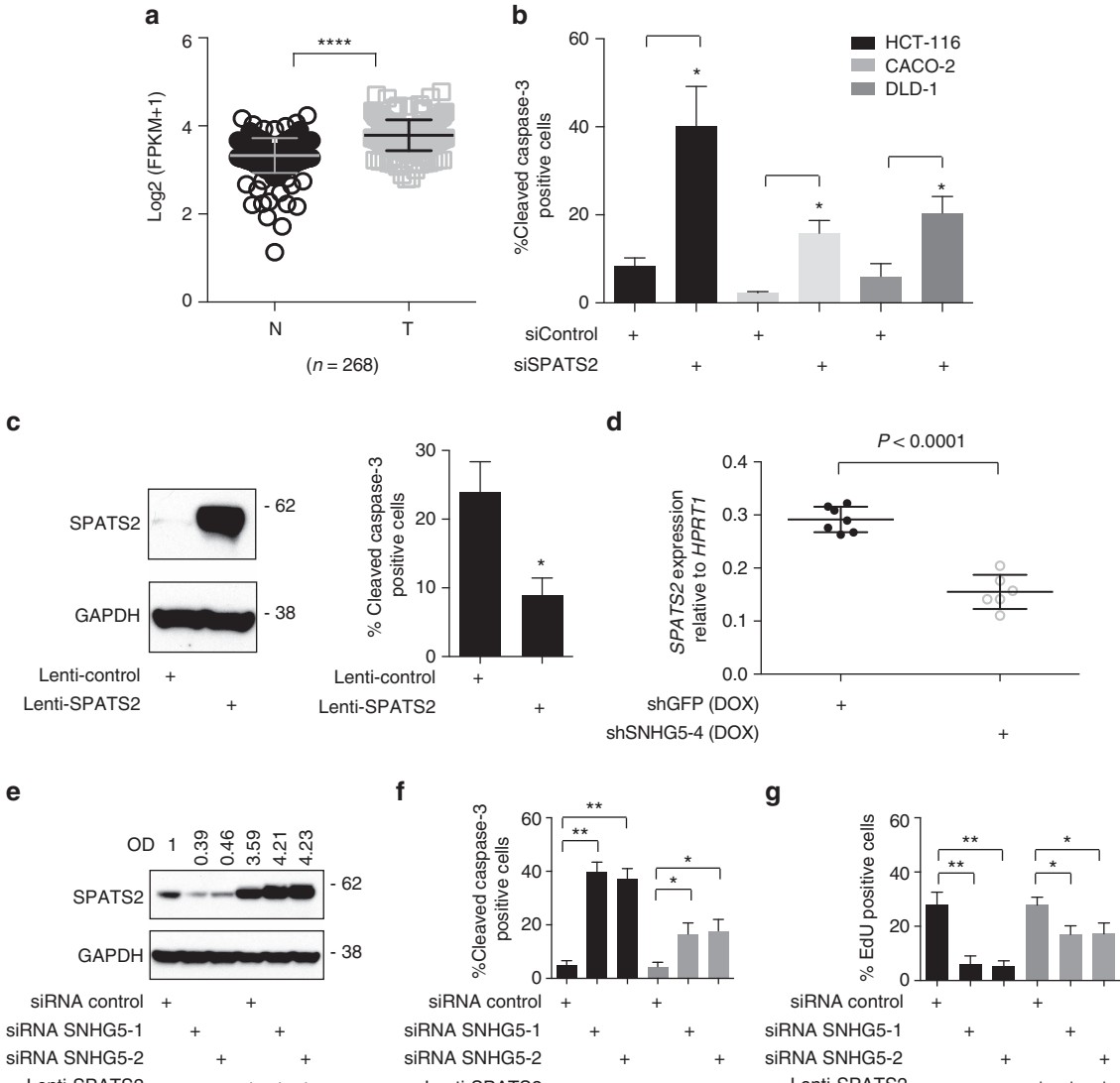

**Figure 5 | The *SNHG5–SPATS2* axis regulates viability in CRC cells.** (**a**) Scatter plot. The expression levels of *SPATS2* were profiled in paired CRC clinical specimens by RNA-seq, where the RNA was extracted from either tumor (C) or normal tissues (N) (****$P$ value < 0.0001, Mann–Whitney $t$-test). (**b**) HCT116, CACO-2 and DLD-1 cell lines were transfected with indicated siRNAs. Thirty-six hours after transfection, cells were labelled with cleaved Caspase-3 antibody and analysed by flow cytometry. Data are shown for the mean of three independent experiments. Error bars indicate ± s.e. (*$P$ value < 0.05). (**c**) Left, western blot. SPATS2 expression levels were measured 48 h after stable transduction of HT-29 cells. GAPDH is included as loading control. Right, HT-29 cells the cells were treated with 25 µg ml$^{-1}$ of oxaliplatin. Twenty-four hours later, the cells were stained with cleaved Caspase-3 antibody and analysed by flow cytometry. Error bars indicate the mean ± s.e. of three independent experiments. (*$P$ value < 0.05). (**d**) qRT–PCR. *SPATS2* mRNA levels in shGFP and shSNHG5-4 xenograft tumours at day 35. Data is plotted relative to the *HPRT1* housekeeping gene (****$P$ value < 0.0001). (**e**) Western blot. HCT116 stable transduced with the indicated lentiviral vectors were transfected with indicated siRNAs. SPATS2 expression levels were measured 36 h after transfection. OD of protein bands is indicated relative to corresponding loading control GAPDH and normalized relative the non-targeting siControl. (**f**) HCT116 cells were transfected as in **e**, stained with cleaved Caspase-3 antibody or (**g**) pulsed with EdU after 36 h to measure the number of EdU-labelled cells representing the S-phase populations and analysed by flow cytometry. Data represents the mean of three independent experiments. Error bars indicate ± s.e. (*$P$ value < 0.05, **$P$ value < 0.01).

up-regulated in CRC (Supplementary Fig. 7a). If the stabilizing effect of *SNHG5* on the *SPATS2* mRNA is important for the survival-promoting functions of *SNHG5*, we would expect loss of *SPATS2* to impact on CRC cell survival. Indeed, knockdown of *SPATS2* in CRC cell lines resulted in a significant reduction of cell viability (Fig. 5b; Supplementary Fig. 7b). Furthermore, phenocopying the effects observed on *SNHG5* depletion, also the inhibition of SPATS2 expression down-regulates the STAT3 pathway, hence confirming the impact of SPATS2 expression on viability in CRC cell lines (Supplementary Fig. 7c). Moreover, ectopic SPATS2 expression in HT-29 cells prevented apoptosis

induction on oxaliplatin treatment, thus recapitulating the *SNHG5* overexpression phenotype (Fig. 5c).

To further explore the interaction between *SNHG5* and *SPATS2*, we investigated the *SNHG5* and *SPATS2* expression patterns in primary CRCs and mouse xenograft tumours (Supplementary Fig. 7d). The correlation found in CRC samples was further confirmed by analyzing *SPATS2* levels in the xenograft tumours, where knockdown of *SNHG5* resulted in a significant decrease in *SPATS2* mRNA levels (Fig. 5d). Finally, we investigated whether restoring *SPATS2* expression in *SNHG5*-depleted CRC cells was sufficient to rescue the

apoptotic phenotype. Indeed, in HCT116 cells ectopic expression of SPATS2-mediated a partial rescue from the anti-proliferative and pro-apoptotic effect following the *SNHG5* depletion (Fig. 5e–g). Altogether, this data suggest that the *SNHG5–SPATS2* interaction is important for promoting CRC cell survival.

***SNHG5* impairs the association of *SPATS2* with STAU1.** We hypothesized that the binding of *SNHG5* promoted *SPATS2* mRNA stabilization via protecting the transcript from post-transcriptional regulators. The RNA-binding protein STAU1 has previously been described to destabilize mRNAs in the cytoplasm[23]. Interestingly, knocking down STAU1 resulted in an up-regulation in both transcript and protein levels of *SPATS2* in HCT116 cells (Fig. 6a). To evaluate the effect of STAU1 depletion on *SPATS2* transcript stability and to compare with the effect resulting from *SNHG5* depletion, HCT116 cells were treated for 5 h with Triptolide following *SNHG5* and STAU1 knockdown (Fig. 6b). As expected, the *SPATS2* mRNA stability was reduced on *SNHG5* depletion, whereas the opposite effect was measured after STAU1 knockdown. To validate *SPATS2* as a direct target of STAU1, we performed RNA immunoprecipitation using a STAU1 antibody after ultraviolet cross-link. Importantly, the *SPATS2* mRNA was enriched in the STAU1 pull-down compared with the IgG control. Moreover, depleting *SNHG5* in HCT116 cells increased the association of *SPATS2* with STAU1. Importantly, this was not observed for another known STAU1 target gene, *ARF1* (Fig. 6c)[13]. This data suggests that *SNHG5* impairs the association *SPATS2* with STAU1. Having confirmed *SPATS2* as a key target of *SNHG5*, and STAU1 as a regulator of *SPATS2*, we speculated if STAU1 could be implicated in CRC cell survival. Towards this, we combined knockdown of STAU1 with oxaliplatin treatment and measured the level of apoptosis. As evident from Supplementary Fig. 7e, loss of STAU1 reduced the level of apoptosis implicating STAU1 in CRC cell survival.

We queried the pathway hypothesizing that if *SNHG5* and STAU1 have opposite functions on *SPATS2* mRNA stability, loss of STAU1 might rescue the apoptosis seen following *SNHG5* loss. Towards this, we combined the knockdown of *SNHG5* and STAU1 and measured the resultant level of apoptosis in HCT116 cells. As evident from Fig. 6d,e, loss of STAU1 expression strongly reduced the apoptosis induction following *SNHG5* depletion. In conclusion, our analyses demonstrate that *SNHG5* promotes CRC cells survival by alleviating STAU1-induced degradation of *SPATS2*.

## Discussion

In this study, we used a customized array platform to profile the expression of lncRNAs in a set of CRC samples and to identify lncRNA expression profiles correlating with disease progression. We identified *SNHG5* as significantly up-regulated in CRC with respect to normal tissues and validated this expression pattern in an independent cohort of 313 CRCs. *SNHG5* belongs to a large family of non-coding genes hosting small RNAs, such as snoRNAs and microRNAs, most often residing in the introns of the host genes. Following excision of the small RNAs, these host gene transcripts in general are unstable[24], and many have been viewed as biologically inert carriers, where the small RNAs are thought to be the biologically relevant players[25]. Noticeable exceptions include host genes such as *GAS5*, *TINCR*, *ncRAN* and *MIR31HG*[21,26–28], all of which have biological effects independent of their hosted small RNAs. *SNHG5* has not previously been characterized in CRC but was reported as a translocation partner to *BCL6* in a patient suffering from diffuse large B-cell lymphoma[29]. *SNHG5* encodes the snoRNAs

U50 and U50′ in introns 4 and 5, respectively. Whereas snoRNAs of the C/D box family are well-known to guide fibrillarin-mediated 2′O methylation of the ribosomal RNA, reduced expression and both somatic and germ line mutations in U50 have been reported from breast and prostate cancer[30,31]. Furthermore, a non-canonical function of the U50 snoRNA in modulating RAS activity was recently described[32]. Although currently unclear, this is likely independent from the *SNHG5* mechanism characterized in this paper.

We find that *SNHG5* expression levels are significantly elevated both in the transition between normal tissue and adenomas and between adenomas and early stage (I) carcinoma. Importantly, suppression of *SNHG5* expression not only exerts a strong anti-proliferative effect and induces apoptosis, impacting key survival pathways such as the STAT3 pathway *in vitro*, but also reduces tumour progression *in vivo*. Moreover, *SNHG5* over-expression prevents apoptosis induction in CRC cell lines treated with chemotherapeutic agents. We further characterize *SNHG5* as a stable, cytoplasmic lncRNA not associated with polysomes. This prompted us to explore mechanisms for *SNHG5* in the cytoplasm by isolating mRNAs interacting with the lncRNA using RIA-seq. This is in line with an increasing number of studies describing post-transcriptional mechanisms of gene expression regulation mediated by cytosolic lncRNAs[22]. We identified 121 significant sites of intermolecular interaction with *SNHG5*, located mainly in *cis*-regulatory sequences (3′ UTRs), and further pinpointed *SPATS2* as an important direct target of *SNHG5* in multiple CRC cell lines. *SPATS2* is significantly up-regulated in CRC patients and knockdown of *SNHG5* destabilizes the *SPATS2* mRNA and significantly reduces SPATS2 protein expression. Vice versa, overexpression of SNHG5 leads to increased *SPATS2* mRNA stability and protein expression. *In vitro* luciferase reporter assays demonstrated that this effect is mediated via direct interaction between a sequence motif in the *SNHG5* exon 1 and the 3′ UTR of *SPATS2*, although we cannot rule out other interactions modes with other targets or in other cellular contexts. Importantly, manipulating the expression levels of SPATS2 in CRC cells phenocopies the cellular viability response following *SNHG5* depletion or overexpression. Finally, restoring the expression of SPATS2 in *SNHG5*-depleted cells resulted in a partial rescue of the apoptotic phenotype induced by the inhibition of lncRNA. Hence, *SPATS2* is a key downstream target of *SNHG5*. Interestingly, we find that the expression levels of *SNHG5* and *SPATS2* display a significant negative correlation in adenomas and xenograft tumours following *SNHG5* knockdown, suggesting a contribution of the *SNHG5–SPATS2* axis to tumorigenic events in CRC.

Several RNA-binding proteins have been described to regulate mRNA stability and decay[33]. Here, we demonstrate in CRC cells that the *SNHG5–SPATS2* interaction impairs *SPATS2*-association with STAU1, a major regulator of the cytoplasmic RNA decay. STAU1 is part of a highly conserved family of double-stranded RNA-binding proteins implicated in mRNA transport, stability and translation; these processes depend on the ability of STAU1 to bind intra- or intermolecular RNA duplexes, most prominently found in 3′ UTRs of target mRNAs, and to recruit components of the cytosolic decay machinery, such as UPF1 (ref. 13). STAU1 is ubiquitously expressed, and whereas an important role for STAU1 in the development of the central nervous system has been described, the involvement of STAU1 in tumorigenic processes remains largely unexplored. STAU1 has previously been linked to lncRNA functions as a post-transcriptional regulator of the stability of specific mRNAs involved in epidermal terminal differentiation[21,34] and cell cycle entry in gastric cancer cell lines[35]. We analysed a recently published hiCLIP dataset for STAU1 (ref. 17) and did not find such sites

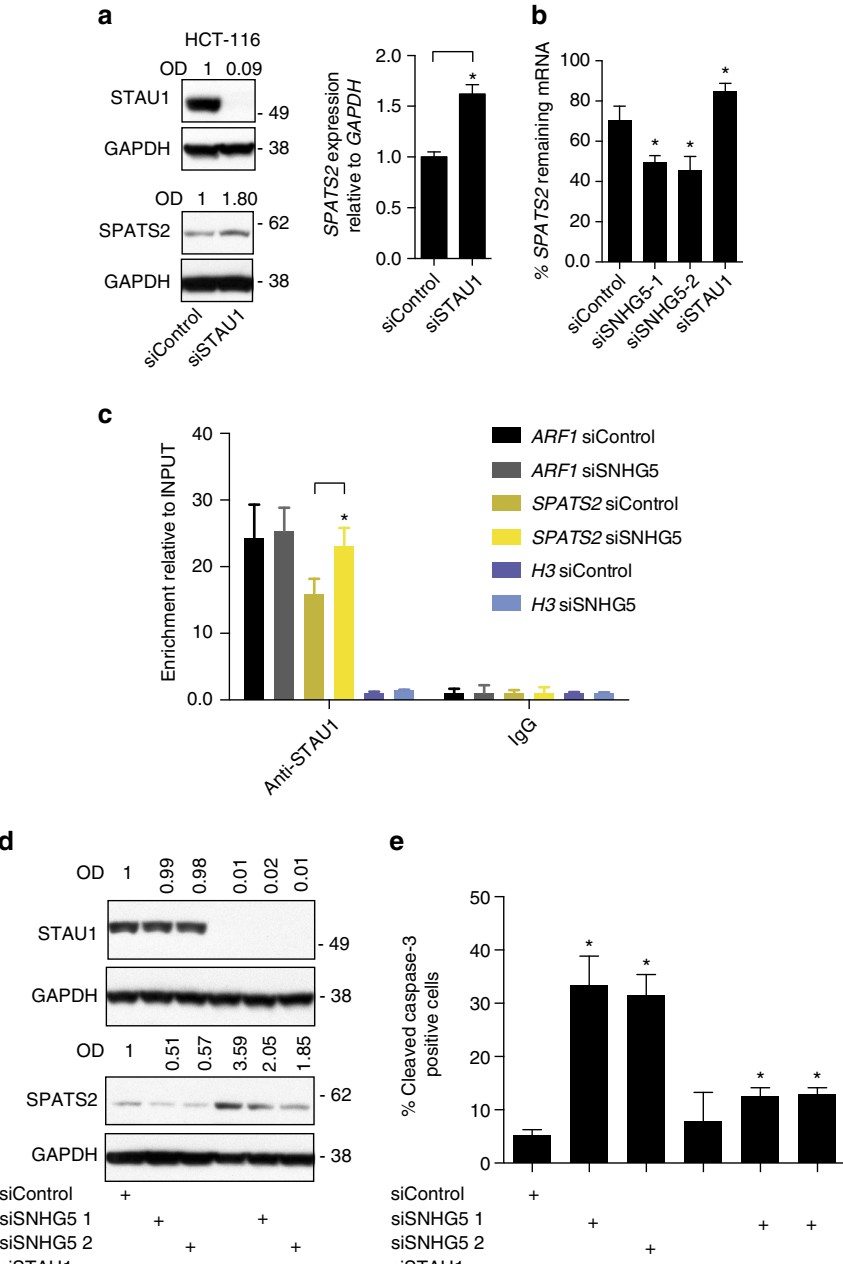

**Figure 6 | *SNHG5* impairs the association between STAU1 protein and the *SPATS2* mRNA.** (**a**) Left, western blot for STAU1 and SPATS2 in HCT116 cells transfected with indicated siRNAs for 72 h. OD of protein bands is indicated relative to corresponding loading control GAPDH and normalized relative the non-targeting siControl. Right, qRT–PCR. HCT116 cells were transfected as described above. *SPATS2* expression data are shown normalized to the *GAPDH* housekeeping gene and normalized to the non-targeting control siRNA. (**b**) qRT–PCR. HCT116 cells were transfected with the indicated siRNAs. Forty-eight hours after transfection the cells were treated with Triptolide at a final concentration of 10 μM for 5 h and the RNA subsequently extracted. The percentage retrieved *SPATS2* mRNA was obtained by normalizing to the corresponding expression levels in the untreated cells. (**c**) RNA immunoprecipitation. HCT116 were transfected with the indicated siRNAs for 48 h. The cytosolic fraction was isolated from the ultraviolet-cross-linked cells and the IP performed with a STAU1-specific antibody. Rabbit IgG were included as negative control for the immunoprecipitation. The RNA was extracted, retro-transcribed and the *SPATS2* mRNA levels evaluated. *ARF1* was included as positive control and H3 mRNA as negative control. Error bars indicate ± s.d. relative for the mean of three independent experiments (*P value < 0.05, stats method). (**d**) Left panel, western blot. HCT116 cells were transfected with indicated siRNAs for 72 h and analysed for STAU1 and SPATS2 using western blotting. OD of protein bands is indicated relative to corresponding loading control GAPDH and normalized relative the non-targeting siControl. Right panel, Flow Cytometry. (**e**) HCT116 cells were transfected as in **d**, the cells stained with cleaved Caspase-3 antibody, and analysed by flow cytometry. Error bars indicating ± s.d. relative for the mean of three independent experiments (*P value < 0.05, **P value < 0.01).

in *SPATS2*. This, however, is unsurprising as that study was performed in HEK293 cells in which *SPATS2* is very lowly expressed. Although we cannot rule out an impact of STAU1 on SPATS2 translation, our data show that *SPATS2* is sensitive to STAU1-mediated mRNA decay and that *SNHG5* can limit the association of STAU1 to *SPATS2*. We also observe that STAU1 knockdown rescues the apoptotic phenotype resulting from *SNHG5* depletion, thus demonstrating a genetic interaction

between the *SNHG5* and STAU1, promoting the stabilization of *SPATS2*. An intriguing hypothesis is that the up-regulation of *SNHG5* during early stages of tumour progression may be required in the transforming cells to buffer the STAU1-mediated destabilization occurring on the *SPATS2* mRNA, as well as other target mRNAs, hence promoting the survival of the pre-cancerous cells. Hence, inhibition of STAU1 decay activity on specific mRNA target provides an intriguing hypothesis for important early steps in CRC development but additional studies are needed to evaluate the overall importance of this mechanism.

Altogether, our data describe a role of *SNHG5* in regulating cell survival in CRC and suggests *SNHG5* as a potential candidate for RNA-based anti-cancer drug studies.

## Methods

**Cell culture and animal experiments.** HCT116, LS174T and HT-29 CRC cell lines were maintained in McCoy's medium (Invitrogen) supplemented with 10% fetal bovine serum (Hyclone) and penicillin/streptomycin (Invitrogen). CACO-2, and SW620 and HEK293 FT cell lines were maintained in DMEM medium (Invitrogen) supplemented with 10% fetal bovine serum and penicillin/streptomycin. DLD-1 cell line was maintained in RPMI medium (Invitrogen) supplemented with 10% fetal bovine serum and penicillin/streptomycin. Cell lines were obtained from the ATCC. DNA plasmids were transfected at a final concentration of $2.5\,\mu g\,ml^{-1}$ by forward transfection using FuGENE HD Transfection Reagent (Promega, Fitchburg, WI, USA) according to manufacturer's procedures. siRNA oligonucleotides (listed in Supplementary Table 3) were transfected at a final concentration of 50 nM by reverse transfection using RNAiMAX (Invitrogen), according to the manufacturer's instructions. Cells were treated with oxaliplatin (EIMC United pharmaceutical) in aqueous solution at indicated concentrations and times, where appropriate. For the RNA stability assay the cells were treated at the indicated time points with $5\,\mu g\,ml^{-1}$ Actinomycin D or $10\,\mu M$ Triptolide (Sigma, St Louis, MO, USA). NMRI-nude adult females were obtained from Janvier Labs. All mouse experiments were approved by Dyreforsøgstilsynet according to Danish legislation (licence number 2012-15-2934-00306).

**Tissue samples preparation and microarray analysis.** The cancer specimens included in this study are from the colorectal cancer biobank at the Department of Molecular Medicine, Aarhus University Hospital. A total of 44 MSS and MSI primary stage I–IV (T2-4, N0-3, M0/1) CRCs, 39 adenomas and 20 normal colon mucosa samples were used in the array analyses. Finally, 280 fresh frozen microsatellite stable (MSS) or microsatellite instable (MSI), primary stage I-IV (T2-4, N0-3, M0/1) CRCs, 33 adenomas and 294 normal colon mucosa were used for validation using RNA-seq. The samples were obtained directly from surgery after removal of the necessary amount of tissue for routine pathology examination. Patients who had received preoperative chemotherapy and/or radiation of rectal cancers were excluded. Postoperatively, the tumours were histologically classified and staged according to the TNM system. Cases with hereditary colorectal cancer syndromes were not included in the study. The use of the human tissue samples for research purpose was approved by the Central Denmark Region Committees on Biomedical Research Ethics (DK; 1999/4678). Informed written consent was given by all participants. RNA from both tissues and cell lines was extracted using the RNeasy Mini kit following the manufacturer's instructions. The microarray platform and analysis was previously described[18]. Arrays were normalized using the RMA implementation of the oligo software package and subsequently analysed using the LIMMA package. For differential transcript expression analysis, a linear model was fitted to the expression data.

**RNA-FISH.** HCT116 and TIG-3 human fibroblast were seeded and reverse transfected with the indicated siRNAs in chamber glass slides (LAB-TEK). Twenty-four hour after transfections, the cells were fixed in 3.7% formaldehyde (cat. 28906, Thermo Scientific, Rockford, IL, USA) overnight. Permeabilization was carried out with 70% EtOH at 2–8 °C for at least 60 min followed by 30 min incubation with 0.1 M triethanolamine (Sigma) + 0.5% (v/v) acetic anhydride (Sigma) in water at room temperature (RT). The slides were prehybridized for 1 h at 55 °C with 100 µl of pre-hybridization buffer (50% formamide (Sigma) + 5 × SSC + 5 × Denhardts solution + 0.5% Tween20 (Thermo scientific)). dFAM-labelled LNA-probes (Exiqon) were added to a final concentration of 20–30 nM in 100 µl of pre-hybridization buffer and incubated at 55 °C for 1 h in a humidified chamber. The slides were washed 3 × 10 min in 0.1 × SSC (Ambion) at 55 °C, followed by 3 × 2 min wash in 1 × PBS (Gibco) at room temperature and subsequently blocked in blocking buffer (10% heat inactivated goat serum (cat. 34864, Sigma) + 0.5% blocking reagent in PBS-Tween) for 1 h at RT. For each slide 150 µl anti-FAM-POD diluted 1:40 in blocking buffer was added and incubated for 1 h at room temperature. The slides were washed twice 2 min in 1 × PBS and 50 µl Cy3-TSA substrate diluted 1:50 in

diluent and incubated in the dark for 10 min at room temperature. Finally, the slides were mounted with DAPI (Life Technologies) nuclear stain. The SNHG5 probe can be obtained from Exiqon.

**Vector construction.** We amplified the entire SNHG5 genomic locus ( ± 200 bp upstream and downstream the locus) by PCR (HiFi Platinum Taq, Invitrogen) using the following primers: gSNHG5 FW 5′-TACTGGCTGCG-CACTTCG-3′, gSNHG5 RV 5′-TACCCTGCACAAACCCGAAA-3′. The amplicon was extracted from gel and cloned in pGEM T-Easy vector system (Promega) and further subcloned into pcDNA 3.1( + ) (Invitrogen). The SNHG5 cDNA sequence was synthetically cloned in pUC57 backbone (Genescript technology) and subcloned into pcDNA 3.1. The pSNHG5 cDNA mutants were obtained using a Quickchange mutagenesis kit (Agilent) following manufacturer's instructions. The primers used for the mutagenesis are listed in Supplementary Table 1. The lentiviral vector for the stable expression of SPATS2 cDNA (PLOHS_ccsbBEn_03998, GE Dharmacon) was used as template for subcloning into pcDNA 3.1( + ) backbone using the following primers: SPATS2 cDNA forward EcoRI 5′-GAATTCGTGGCTCTAGAAGGGGAG GTGG-3′; SPATS2 cDNA reverse EcoRI 5′-GAATTCAATTAAATGC-TACTTTCTCATAG-3′.

**Crystal violet staining.** Cells were seeded and reverse transfected in 6-well plates. Twenty-four, forty-eight and seventy-two hour after transfection, cells were washed twice in PBS (Gibco) and fixed with 4% formalin (Sigma) for 10 min, and stained with 0.1% crystal violet solution (Sigma). The amount of crystal violet staining was quantified using the 'ColonyArea' ImageJ plugin (Promega).

**EdU incorporation assays.** Cells grown in 6-well plates (NUNC) under appropriated conditions were pulsed with EdU (33 µM) for 20 min before fixation. The EdU was detected using Click-IT EdU Alexa-Fluor 647 Flow Cytometry Assay Kit (Molecular Probes, Life Technologies), according to the manufacturer's protocol. Data were obtained by flow cytometric analysis of 15,000 cells per treatment and analysed with the FlowJo 887 software (Tree Star, Ashland, OR, USA).

**Cleaved Caspase-3 staining.** Cells were reverse transfected using RNAiMax reagent (Invitrogen) in six-well plates (NUNC). 36 h after transfection the cells were harvested and fixed in 4% formaldehyde methanol-free (Thermo Scientific) and incubated for 30 min on ice in 90% methanol in 1 × PBS pH 7.4 (Gibco). The cells were first stained with anti-cleaved Caspase-3 antibody (#9664, Cell signaling) in 1 × PBS-2% BSA (Sigma) for 1 h at room temperature in the dark, washed twice in 1 × PBS-2% BSA and stained with Alexa-Fluor 647 goat anti-rabbit secondary antibody (Life technology) for one additional hour at room temperature protected from light. For propidium iodide staining, the labelled cells were incubated in propidium iodide (Sigma) and RNase-A (Sigma) solution for 30 min at 37 °C. Data analysis was performed using the FlowJo 887 software.

**Western blot analysis.** Cells were seeded and reverse transfected in 6-well plates (NUNC). After 36 h, cells were harvested, washed once with PBS and the pellets lysed in RIPA buffer (150 nM NaCl, 0.1% sodium deoxycholate (Sigma), 0.1% SDS (Sigma), 50 mM Tris–HCl (pH 8), 1 mM EDTA (Sigma)) containing protease inhibitors (Complete Mini Protease Inhibitor Cocktail; Roche Applied Science). Proteins were separated by electrophoresis in 4–12% NuPAGE Bis–Tris gels (Invitrogen) and transferred to nitrocellulose membranes. The primary antibodies used were: GAPDH (Santa Cruz, sc-25778), cleaved PARP-1 (Cell signaling, 5625), PITRM1 (Pierce, PIEAPA531558), GLE1 (Abcam, ab69968), SPATS2 (Santa Cruz, sc-390306), Bcl-xL (Cell Signaling, 2762), Bcl-2 (Cell Signaling, 2876), STAT3 (Cell Signaling, 9132) and STAU1 (Proteintech, 14225-1-AP). Un-cropped scans of all western blots presented in the main figures are presented in Supplementary Fig. 8.

**Northern blot.** Cells were seeded and reverse transfected in 6-well plates (NUNC). After 36 h total RNA was isolated using Trizol reagent (Invitrogen). The RNA was separated by electrophoresis in 10% TBE-Urea gel (Acrylamide 10%, TBE-Urea 7M, APS 1 × , Temed 1 ×) and transferred to nitrocellulose membranes (Amersham). The membrane was cross-linked once at 120,000 µJ cm$^{-2}$ with Stratalinker UV Crosslinker (Stratagene). Pre-hybridization was carried out at 50 °C for 1 h in Rapid hyb buffer (Amersham). The DNA probes were $^{32}$P-labelled with T4 PNK kinase (New England Biolabs) and the hybridization carried out 1.5 h at 50 °C. The membrane were 20 min in 50 ml 5 × SSC, 0.1% (w/v) SDS at room temperature and 2 × 15 min in 50 ml 1 − 0.1 × SSC, 0.1% (w/v) SDS at 50 °C. The probe signal was detected using FujiFilm Fluorescent Image Analyzer FLA-3000. DNA probes used were as follows:

U50 5′-TATCTCAGAAGCCAGATCCG-3′; The RNU6B probe was obtained from Exiqon.

**Cell fractionation.** Cells were grown in 15 cm dishes (NUNC). For each sample 1 million cells were harvested and the nuclear/cytoplasmic fractionation was performed using Nuclei EZ Lysis Buffer (Sigma), following the manufacturer's protocol. In the *SNHG5* siRNA pool knockdown experiments HCT116 cells were harvested 36 h after transfection using RNAiMax (Invitrogen).

**RNA extraction and qRT–PCR analysis.** Total RNA was isolated using Trizol reagent (Invitrogen), treated with TURBO DNase (Ambion, Life Technologies) and reverse transcribed using TaqMan Reverse Transcription kit (Applied Biosystems) using random hexamer primers. QRT–PCRs were performed using Sybr Green PCR Fast PCR Master Mix 2× (Applied Biosystems). The house-keeping genes *RPLP0*, *HPRT1* and *GAPDH* were used for normalization of qRT–PCR data, unless otherwise stated. The primers used are listed in Supplementary Table 1.

**RNA sequencing.** HCT116 cells were reverse transfected with All-star negative control (Qiagen), siSNHG5-1 or siSNHG5-2 for 36 h in 6-well plates (NUNC). RNA was extracted from three biological replicates, the quality assessed on the 2100 expert Bioanalyzer (Agilent) and sent for library preparation and sequencing on the Illumina Hiseq2000 by BGI Genomics (Hong Kong).

**In vitro transcription and copies number quantification.** Templates containing the T7 promoter were generated by PCR with high-fidelity DNA polymerase Platinum Taq (Invitrogen). PCR products were purified from agarose gels and 250 ng of template was *in vitro* transcribed using T7 RNA polymerase and the MAXIscript kit (Ambion, Life Technologies) for overnight at 37 °C. After 30 min DNAse treatment with TURBO DNA free at 37 C, the reactions were stopped by adding 0.5 mM EDTA. The concentration of the transcript was measured by A260 values and converted to the number of copies using the molecular weight of the RNA. Dilutions of this transcript were retro-transcribed as described above and the complementary DNA was used as standard curve for qPCR. The primers used for qPCR are: *SPATS2* T7 forward 5′-GGATCCTAATACGACTCACTATAGTAGAAGGGGAG GTGGAGGAT-3′; *SPATS2* T7 reverse 5′-GGATCCGCATTGGGCATTTTG AAGAA-3′; *SNHG5* T7 forward 5′-GGATCCTAATACGACTCACTATA GCGGGTGGTAGGAACAATGG-3′; *SNHG5* T7 reverse 5′-GGATCCTTT CATGTTTGTAAAACGAAGAGC-3′.

**Luciferase assay.** Luciferase reporter constructs containing regions of the *SPA TS2* (NM_001293286.1) and *GLE1* (NM_001003722) 3′ UTRs were cloned into pMIR-REPORT *firefly* luciferase vector (Ambion). pRL-Tk Renilla luciferase reporter was used as transfection control and luciferase assay normalization. The assays were performed 48 h after transfection of the indicated constructs into 2.4 × 10⁴ HEK293 cells per well (four wells each samples) seeded in 96-well plates. The cells were transfected with 50 ng of firefly luciferase vectors and 1 ng of the pRL-Tk Renilla reporter. The firefly reporters were competed by co-transfecting 50 ng of pcDNA 3.1 (+) expressing the SPATS2 cDNA. The reporter activities were measured using the Dual Glo Luciferase assay System (Promega) and GloMax Multi Detection System (Promega). The following primers were used for the fragment subcloning PmeI–HinIII into pMIR-REPORT: *SPATS2* 3′ UTR forward PmeI primer 5′-GTTTAAACGCCTCTATCCCAGAATGTGC-3′ and reverse HindIII primer 5′-AAGCTTTGCTACTTTCTCATAGACTTCCCTA-3′; *GLE1* 3′ UTR forward PmeI primer 5′-GTTTAAACCCTCCCTTGTCTC-TAGTGTCTT-3′ and reverse Hind III primer 5′-AAGCTTCAACCATAACCCA-GACTTGCAT-3′.

**In vivo imaging acquisition to follow tumour development.** A luminescent variant of the HCT116 cell lines were generated by transduction with the pFULT lentiviral vector containing the luciferase2-tdTomato cassette as described before[36]. The shRNA expressing cassettes harbouring a Tet-inducible promoter were generated as described previously by other labs[37]. Two millions freshly harvested cells from these shRNA subclones (200 µl from a stock solution of 10 millions cells per ml of PBS) were subcutaneously injected into the right flank of 22 weeks old NMRI-nude adult females (Janvier Labs). Tumour growth was measured based on the luminescent signal using the IVIS Lumina II system (Perkin Elmer), every 2–4 days from 1 week after cells injection. Mice were treated with doxycycline (*ad libitum*, active compound diluted in drinking water at 2.5 mg ml⁻¹, from Piggidox powder, Boehringer Ingelheim) when shGFP tumours reached an average of 100 mm³. The doxycycline solution was changed every 2–3 days until the end of the experiment. The shRNA sequences are listed in Supplementary Table 3.

**RNA interactome analysis and sequencing.** HCT116 cells were collected (30 million per sample) and resuspended in 20 ml of PBS containing 1% Glutaraldehyde (Sigma) for 10 min at room temperature rotating. Glutaraldehyde was quenched with 2 ml of 1.25 M Glycine (Sigma) (1/10 total volume) for 5 min. The cell pellets were lysed in 2 ml lysis buffer (Tris–Cl pH 7 50 mM,

EDTA RNAse-free 10 mM, SDS 1%, PMSF 1 ×, protease inhibitor cocktail, Superase-In (Invitrogen)) and sonicated four times (25 cycles each, high pulse 30″ ON–45″OFF) using a BIORUPTOR (Diagenode). Cell lysates were centrifuged for 20 min @at 4 °C × 16,100g and the supernatant aliquoted for the immuno-precipitation. The lysates were pre-cleared for 30 min at 4 °C with 50 µl Dyna-beadsMyOne Streptavidin C1 (Life Technologies) in 2 ml of Hybridization solution (NaCl 750 mM, SDS 1%, Tris–Cl pH 7 50 mM, EDTA 1 mM, formamide 15%, PMSF 1 ×, protease inhibitor cocktail, Superase-In, yeast RNA (Ambion) 0.1 mg ml⁻¹). One ml of hybridization buffer containing 100 pmol of probes per ml of lysate was added to each sample and incubated for 4 h at 37 °C rotating. In parallel, the beads for the immunoprecipitation were blocked in hybridization solution 30 min at 4 °C. We added 100 µl of beads per 100 pmol of probes and incubated rotating 30 min at 37 °C. The samples were washed five times (5 min each) in 1 ml washing buffer (2× SSC, SDS 0.5%, PMSF 1 ×) at 37 °C rotating. Each sample was Proteinase K (Invitrogen) treated for 45 min at 50 °C shaking before RNA was extracted. A total of 150 ng from each sample were sent for library preparation and sequencing on the Illumina Hiseq2000 at BGI Genomics (Hong Kong) using Truseq protocol. Primers and biotinylated probes sequences used for the pull-down and experiment validation are listed in Supplementary Tables 2 and 3, respectively.

**RIA-seq bioinformatics analysis.** *Trimming.* The short reads were trimmed using cutadapt to remove errors such as, for example, trailing adapter sequences. The following deviations from default were used: error-rate = 0.05, minimum-length = 25, overlap = 4, quality-cutoff = 20 as well as a list with the relevant Illumina adapter sequences given to the '-a' option.

*Mapping.* Because the reads were generated from a mixture containing post-spliced mRNA, it was necessary to use a mapping approach that takes splice junctions into account. We used the STAR aligner with default options except for outFilterMismatchNoverLmax = 0.1 and outFilterMatchNmin = 16.

*Peak calling.* Peak calling was accomplished using the MACS2 algorithm with the following non-default options: $P = 1e - 3$, to-large, nomodel and shiftsize = X, where 'X' (the peak shift) was estimated individually for each sample using the SPP programme. The fLUC samples were pooled and used as reference background. The IDR algorithm was used to post-process the MACS2 results and to estimate the number of reproducible peaks from the two ODD sample replicates. Owing to a failed replicate, however, the IDR algorithm could not be applied to the EVEN condition, and instead the numbers of reproducible peaks for EVEN were extrapolated from the ODD condition. This gave a conservative result of 736 peaks and an optimal result of 905 peaks for both conditions, although any peaks called in intron areas were then subsequently removed, as these are most likely errors arising from an incorrect peak shift.

*Motif finding.* We used the meme-chip programme for motif finding. As input for meme-chip we used a region of 250 bp flanking each peak summit, for a total of 501 bp long sequences extracted from both ODD and EVEN peak lists. To prevent biases between the sizes of the two sets, sequences were randomly extracted from the larger set (EVEN) until the size of the smaller (ODD) was reached. Default parameters were used except that '-norc' was used. This option prevents the programme from searching the reverse complement, which is irrelevant here as the sequences under consideration are single-stranded RNA.

**RNA immunoprecipitation.** HCT116 cells were cross-linked using 1500 µJ cm⁻² and collected in cold PBS. Fifty millions cells per samples were subjected to nuclear/cytoplasmic fractionation using Nuclei EZ Lysis Buffer (Sigma), cyto-plasmic fractions were collected and the volume adjusted in 2× polysome buffer (Tris–Cl pH 7.4 20 mM, NaCl 200 mM, MgCl₂ 2.5 mM, Triton-X 1%, DTT 1 mM, protease inhibitor cocktail, Superase-In). Lysates were pre-cleared with yeast RNA (0.1 mg ml⁻¹) and recombinant protein G agarose (Thermo Scientific) for 30 min at 4 °C. One per cent of the sample was used as input, the remaining samples incubated with STAU1 antibody (Proteintech, 14225-1-AP) or IgG from rabbit serum (I5006, Sigma) overnight at 4 °C, The RNA-antibody complexes were collected by incubation with Dynabeads protein G beads (Invitrogen). The samples were washed four times and Proteinase K treated before the RNA was extracted for qRT–PCR analysis.

**RNA–RNA interaction prediction.** Thermodynamic stable interaction sites of *SNHG5* (ENST00000414002), the 3′ UTR of *SPATS2* (NM 001293285.1) and the 3′ UTR of *GLE1* (NM 001003722.1) were predicted with IntaRNA (version 1.2.5; default parameters)[38]. The interaction sites of lowest energy with *SPATS2* (− 15.50 kcal mol⁻¹) and *GLE1* (− 14.7 kcal mol⁻¹) span the exons 1 and 2 of SNHG5. The conservation of the interaction site was tested with PETcofold (version 3.2; maximal percentage of gaps in alignment column -g is set to 0.5)[39]. The transcript alignments were extracted from the genome-wide hg19 46way multiz alignments. We used the 'Stitch MAF blocks' function of the Galaxy browser to get a connected alignment for each exon, removed all sequences with more than 50% gaps from the alignments, and concatenated exons of the species that occurred in all exons.

**Statistical testing.** All *P* values are calculated using a two-tailed Student's *t*-test, unless otherwise stated. For the analysis of the growth curves from the xenograft experiment, the following method was applied: Categoric variables were reported by means of contingency tables. Furthermore, for continuous variables, median and range were computed. To investigate their associations with the product and group parameters, univariate statistical analyses were performed using Kruskal–Wallis or Mann–Whitney non-parametric test. More-over, multivariate analyses were carried out. Linear mixed-regression model, containing both fixed and random effects, was used to determine the relationship between luminescence growth and number of days after injection. The fixed part of the model included variables corresponding to the number of post-injection days and the different product treatments. Interaction terms were built into the model; random intercepts and random slopes were included to take time into account. The coefficients of the model were estimated by maximum likelihood and considered significant at the 0.05 levels. The power of analysis was reduced because of the limited number of xenograft tumours and few tumours did not have measurements for all variables. All *P* values reported are two sided and the significance level was set at 5% ($P < 0.05$) Statistical analysis was performed using the STATA 13 software (Stata Corporation, College Station, TX, USA).

**Data availability.** The datasets generated during and/or analysed during the current study are available: the CRC microarray data: GSE76713. The CRC RNA-seq data: Supplementary Table 4. The RNA-seq data from HCT116 cells with knockdown of SNHG5: GSE76801. The RIA-seq data from HCT116 cells for SNHG5: GSE76792.

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

## Acknowledgements

We thank Disa E. Tehler and Sophia J. Häfner for commenting on the manuscript; Kim Holmstrøm (Bioneer) for help with the RNA *in situ* hybridization experiments, and Steen Holmberg and Darima Petersen for help with the polysome profile experiments (Copenhagen University, DK). We thank Pr. Sanjiv Sam Gambhir (Stanford University, USA) for providing the pFULT lentiviral vector. Work in the Lund laboratory was supported by funding from the People Programme (Marie Curie Actions) of the European Union's Seventh Framework Programme FP7/2007-2013/ under REA Grant Agreement 607720, the Danish Council for Independent Research (Sapere Aude programme), the Novo Nordisk Foundation, the Lundbeck Foundation, the Danish National Research Foundation (DNRF82) and the Danish Cancer Society.

## Author contributions

N.D.D. and A.H.L. designed the experiments, analysed the data and wrote the manuscript. All coauthors commented on the manuscript. N.D.D., M.M., C.C. and H.M.G. conducted the experiments. M.M.N., C.F.R., S.S., N.R., S.V. S.T. and J.S.P. performed bioinformatics analyses. L.L.C., T.Ø. and C.L.A. contributed clinical samples and analyses hereof, and R.B. contributed technical expertise on the RIA-seq protocol. All authors commented on the manuscript.

## Additional information

**Competing financial interests:** The authors declare no competing financial interests.

**Publishers's note:** 

