## [Peer Review File · Nature Communications]

Reviewer #1 (Remarks to the Author)

In this interesting study, Lund and co-workers identify SNHG5 as a cytoplasmic long non coding RNA with increased expression in colorectal cancer. Knock-down of SNHG5 induces apoptosis while overexpression protect cancer cells from oxaliplatin-induced apoptosis. Furthermore, in mice injected cancer cells do not survive when SNHG5 is down-regulated. To identify RNA targets of SNHG5, gene analysis upon downregulation of SNHG5 was performed. This resulted in 150 deregulated genes. Additionally, RIA-seq further confirmed their gene analysis results. Interestingly, SNHG5 stabilizes target RNAs and, therefore, increases expression level of encoded proteins. The authors then focus on one of the identified targets, SPATS2, which is claimed to be a Stau1 target. In a preliminary set of experiments, the authors provide evidence that (i) down-regulation of Stau1 might increase SPATS2 at the protein as well as mRNA level; (ii) that SPATS2 is detected in Stau1 IPs and finally, (iii) that down-regulation of SNHG5 further enhances SPATS2 levels in the Stau1 IP.

General comments

In general, the finding that the long non coding RNA promotes colorectal cancer cell survival is of high interest to the general public, the experiments are well performed and the quality of the first four figures is convincing. Unfortunately, the latter part describing a potential link to the RNA binding protein Stau1 and a role in Stau1-dependent mRNA destabilization is preliminary, incomplete and not (yet) convincing. Consequently, the title of the study must be changed and the manuscript cannot be published in its current form.

There are two options for the authors to revise their interesting study. In my opinion, they could simply omit Figure 5 and replace this preliminary and incomplete part with a new figure in which the authors investigate which of the SNHG5 targets changes in colorectal cancer patients. This would significantly strengthen their key findings and provide first molecular insight into the underlying mechanism. Alternatively, the authors would have to invest significantly more time to improve existing figures (especially the IP data in Fig. 5B) and provide more insight into a potential link between SNHG5 and SPATS2 on one side and the role of Stau1 in mRNA destabilization on the other.

The main problem of the Fig. 5 is panel C, as the IP is not (yet) convincing. In addition, the experiment presented does not show that Stau1 directly binds either SNHG5 or SPATS2. The current manuscript makes strong claims about SNHG5 regulating Stau1-mediated mRNA decay. Surprisingly, however, the authors do not provide any RNA levels, only proteins levels (not quantified, see below) are shown. To make a convincing case, the authors have to provide convincing experimental insight whether (i) Stau1 indeed binds SPATS2, (ii) binding is competed by lncRNA and whether (ii) it mediates direct destabilization. Here, they have to relate to recent published work by Jernej Ule in Nature (Sugimoto et al., 2015) and to Melissa Moore in Nature SMB (Ricci et al., 2014: this paper corrects previous findings by another lab that Stau1 actually destabilizes transcripts!).

In the Sugimoto paper, Ule and co-workers showed that although Arf1 mRNA contained hiCLIP sites for Stau1, there was neither destabilization of Arf1 nor for other RNAs genome-wide (actually, Xbp1 mRNA was one of the few that did). Currently, Sugimoto et al. and Ricci et al. convincingly show that there is little evidence that Stau1 actually destabilizes mRNAs as originally claimed. Therefore, it would be important to reanalyse whether SPATS2 indeed has hiCLIP sites for Stau1 and whether there is significant Stau1-decay at the RNA level. Of course, there are additional experiments to consider: when SNHG5 is targeting SPATS2, overexpression of another SNHG5 target should compete with SPATS2 resulting in Stau1 mediated degradation.

In the remainder, there are additional comments for the authors to improve the current manuscript for revision.

1. There is a serious disconnect between the two pools of identified SNHG5 targets depending on which siRNA had been used, as there's only a weak overlap. Does the weak overlap suggest off-target effects? What about the targets identified by RIA-seq (Fig. 3C). What is the overlap between transcripts deregulated upon SNHG5 downregulation and RIA-seq results? And finally, why have

the authors selected those 3 top candidates although they started with 121 transcripts? What were the selection criteria?

2. Have the authors investigated whether the two siRNA oligos exert an additive effect when added together? This would be important to know in the context of several figures in the main manuscript.

3. In general, the authors need to provide more experimental detail for their experiments, e.g. explain siRNA sequences, explain even and odd antisense oligos, be more consistent with figure codes in figures (Fig. 3A versus Supp. Fig. 5a), be more specific in figure legends and methods on describing experiments.

4. I would favour moving the GO term analysis from supplements to Fig. 1.

5. Fig. 2a: Labelling is wrong (symbols for siControl, and siSNG52, triangle and circle), the figure legend 2bc reference for transfection to 2a and not 2b.

6. In Fig. 2b, the authors reference by mistake "cells were transfected as in b" instead of "as in a"

7. The authors should consider changing the order of panels in Fig. 4, as actually in the text, they use the order Fig. 4dc instead of Fig. 4cd.

8. In several occasions, key data need to be quantified, e.g. Fig. 4A: Fig. 5, panels A and D and then compare whether the effects are indeed comparable and consistent. Possibly, the effect on the protein level might be stronger than on the RNA level (Fig. 5A).

9. Fig. 3b: there is a problem of scaling, as the right side is lacking any sign of GAPDH bars.

10. Fig. 4e: are the authors claiming that there is a correlation between SPATS2 vs SNHG5 as the r value is 0.373? This is not convincing. I would suggest omitting this panel or move into supplements.

11. P. 11: Fig. 4 is actually Fig. 4 g

Reviewer #2 (Remarks to the Author)

In this manuscript Damas and co-authors describe their work on lncRNAs in CRC and specifically the SNHG5 which is highly upregulated in colon cancer lines. They find that its expression is crucial for cell cycle progression and protection from apoptosis and knockdown induce cell death and decreased in vitro and in vivo growth. Using RNAseq they identified 121 associated mRNAs. Further validation revealed that SPATS2 is a target and its mRNA is stabilized by interaction with SNHG5. Indeed KD of SNHG5 results in depletion of SPATS2 in a STAU1 dependent fashion.

Although these data are of interest several points limit enthusiasm and require additional experimentation.

1. It is completely unclear how regulation of SPATS2, a protein with poorly studied characteristics, would drive apoptosis and or cell cycle arrest. In the first figure BCL2/XL are denoted as targets of SNHG5. The same holds for STAT3. What is their role in the process? Are these regulated by siSPATS2? The authors should provide more insight in the steps downstream of SPATS2

2. to claim that SPATS2 is a crucial target requires more extensive proof. Is overexpressing SPATS2 protecting from siSNHG5 KD? This is needed to make this point.

3. Also the effects of siSNHG5 on SPATS2 in CACO-2 seem rather limited both on cell cycle and cell death. The authors should explain this more carefully and study KD in other CRC lines to confirm their claims. Is it potentially associated with MSI?

4. KD should also be quantified in the different figures.

Reviewer #3 (Remarks to the Author)

The Lund laboratory has identified SNHG5 as a cytoplasmic long noncoding RNA expressed more abundantly in colorectal cancer cells. The authors show that SNHG5 associates with a number of mRNAs in cells, including SPATS2, PITRM1 and GLE1 mRNAs, and propose that it renders them

stable by competing for the binding of the decay-promoting RNA binding protein Staufen (STAU). The topic of this manuscript is interesting, timely, and novel. However, some of the data need to be strengthened, particularly the data pertaining to the involvement of STAU.

Main comments

1. Introduction: The authors state that the only cytoplasmic lncRNAs described are ceRNAs, but in fact this characterization is incomplete. They need to mention other examples of cytoplasmic lncRNAs, including other examples of lncRNAs associating with mRNAs described in the literature.

2. How many copies of SNHG5 are there per cell? How does this number compare with the numbers of SPATS2, PITRM1 and GLE1 mRNAs?

3. The interaction between SNHG5 and SPATS2, PITRM1 and GLE1 mRNAs needs to be tested in vitro. What are the regions of complementarity that mediate the interaction of SNHG5 with PITRM1 and GLE1 mRNAs? These should be identified as well. The SNHG5 mutant lacking the region of interaction with SPATS2 mRNA should be tested in vitro and in vivo. Does the mutant affect tumorigenesis?

4. The data in Figure 5 need to be developed further:

- the authors do not explain why they decided to focus on STAU and not on many other RNA binding proteins that can promote mRNA decay. Do other such proteins affect SNHG5 function? In fact, they could also have pursued microRNAs to explain this effect. They should explain why STAU was chosen.

- the authors show that STAU binds SPATS2 mRNA, but because the IgG values are not adjusted, it is difficult to see the consequences of STAU binding to SPATS2 mRNA after SNHG5 silencing. The levels of mRNAs in IgG IP (after normalization to a nonspecific RNA) should be adjusted to 1, so that the enrichments in STAU IP samples can be compared. Does the interaction of STAU with SPATS2 mRNA increase when SNHG5 is overexpressed? Does it remain unchanged when the SNHG5 deletion mutant is overexpressed?

- What region of SPATS2 mRNA does STAU bind to? Is the site of interaction of SPATS2 mRNA with the SNHG5 the same as the site of interaction with STAU?

- Does STAU also bind PITRM1 and GLE1 mRNAs? Are these interactions antagonized in a similar manner by SNHG5?

Minor comments

Fig S2e: In the Y axis, please fix a typo (it is 'Retrieved' not 'Retreived')

Please use HCT116 consistently (HCT-116 is less commonly used)

The authors need to adopt the appropriate nomenclature for human and mouse genes, lncRNAs, mRNAs, and proteins.

Rebuttal letter

Reviewers' comments:

Reviewer #1 (Remarks to the Author):Expert in mRNA regulation and STAU

In this interesting study, Lund and co-workers identify SNHG5 as a cytoplasmic long non coding RNA with increased expression in colorectal cancer. Knock-down of SNHG5 induces apoptosis while overexpression protect cancer cells from oxaliplatin-induced apoptosis. Furthermore, in mice injected cancer cells do not survive when SNHG5 is down-regulated. To identify RNA targets of SNHG5, gene analysis upon downregulation of SNHG5 was performed. This resulted in 150 deregulated genes. Additionally, RIA-seq further confirmed their gene analysis results. Interestingly, SNHG5 stabilizes target RNAs and, therefore, increases expression level of encoded proteins. The authors then focus on one of the identified targets, SPATS2, which is claimed to be a Stau1 target. In a preliminary set of experiments, the authors provide evidence that (i) down-regulation of Stau1 might increase SPATS2 at the protein as well as mRNA level; (ii) that SPATS2 is detected in Stau1 IPs and finally, (iii) that down-regulation of SNHG5 further enhances SPATS2 levels in the Stau1 IP.

General comments

In general, the finding that the long non coding RNA promotes colorectal cancer cell survival is of high interest to the general public, the experiments are well performed and the quality of the first four figures is convincing. Unfortunately, the latter part describing a potential link to the RNA binding protein Stau1 and a role in Stau1-dependent mRNA destabilization is preliminary, incomplete and not (yet) convincing. Consequently, the title of the study must be changed and the manuscript cannot be published in its current form.

There are two options for the authors to revise their interesting study. In my opinion, they could simply omit Figure 5 and replace this preliminary and incomplete part with a new figure in which the authors investigate which of the SNHG5 targets changes in colorectal cancer patients. This would significantly strengthen their key findings and provide first molecular insight into the underlying mechanism. Alternatively, the authors would have to invest significantly more time to improve existing figures (especially the IP data in Fig. 5B) and provide more insight into a potential link between SNHG5 and SPATS2 on one side and the role of Stau1 in mRNA destabilization on the other.

The main problem of the Fig. 5 is panel C, as the IP is not (yet) convincing. In addition, the experiment presented does not show that Stau1 directly binds either SNHG5 or SPATS2.

➤ Author response:

Thank you for the positive remarks and constructive suggestions. In response we now provide data on the expression patterns of the

SNHG5 targets *SPATS2*, *PITRM1* and *GLE1* in CRC. This data is presented in Fig. 5a and Fig S7. In accordance with our hypothesis, all 3 *SNHG5* targets are indeed up-regulated in CRC.

Regarding the mechanism of *SNHG5* and the importance of *SPATS2* and *STAU1*, we focused our efforts on demonstrating a functional importance of *SPATS2* downstream *SNHG5*. Towards this, we performed rescue experiments and examined the direct interaction between *SNHG5* and *SPATS2*. As can be seen from Fig. 5, overexpression of *SPATS2* partially rescues the apoptotic phenotype resulting from *SNHG5* depletion, thus demonstrating *SPATS2* as an important downstream target for *SNHG5*. Furthermore, we created luciferase reporter vectors holding 3'UTR fragments from *SPATS2* and *GLE1* and demonstrate in Fig. 4 and Fig. S6, that these are responsive to *SNHG5* and that this responsiveness as expected is dependent on exon 1 of *SNHG5*.

Regarding the *STAU1* IP presented in the original Fig. 5c (now Fig. 6C) we do not agree with Reviewer 1. These experiments were performed using UV cross-linking (not formaldehyde) and thus do demonstrate a direct interaction between *STAU1* and *SPATS2*. We have no evidence that *STAU1* binds directly to *SNHG5*, which we would also not expect if *SNHG5* competes with *STAU1* for binding to *SPATS2*. The direct interaction between *SNHG5* and *SPATS2* was addressed using reporter vectors as described above.

The current manuscript makes strong claims about *SNHG5* regulating *Stau1*-mediated mRNA decay. Surprisingly, however, the authors do not provide any RNA levels, only proteins levels (not quantified, see below) are shown. To make a convincing case, the authors have to provide convincing experimental insight whether: (i) *Stau1* indeed binds *SPATS2*, (ii) binding is competed by lncRNA and whether (iii) it mediates direct destabilization.

Here, they have to relate to recent published work by Jernej Ule in Nature (Sugimoto et al., 2015) and to Melissa Moore in Nature SMB (Ricci et al., 2014: this paper corrects previous findings by another lab that *Stau1* actually destabilizes transcripts!).

In the Sugimoto paper, Ule and co-workers showed that although *Arf1* mRNA contained hiCLIP sites for *Stau1*, there was neither destabilization of *Arf1* nor for other RNAs genome-wide (actually, *Xbp1* mRNA was one of the few that did). Currently, Sugimoto et al. and Ricci et al. convincingly show that there is little evidence that *Stau1* actually destabilizes mRNAs as originally claimed. Therefore, it would be important to reanalyse whether *SPATS2* indeed has hiCLIP sites for *Stau1* and whether there is significant *Stau1*-decay at the RNA level. Of course, there are additional experiments to consider: when *SNHG5* is targeting *SPATS2*, overexpression of another *SNHG5* target should compete with *SPATS2* resulting in *Stau1* mediated degradation.

➤ Author response:

- (i) Regarding the concern that STAU1 may not bind directly to *SPATS2*, this would indeed have been valid if the IPs had been performed from formaldehyde treated samples. However, as stated above, Fig. 6C demonstrates that STAU1 binds *SPATS2* in UV treated samples, which would require a direct interaction.
- (ii) As can be seen in Fig. 6C, depletion of *SNHG5* leads to an increased association of *SPATS2* to STAU1.
- (iii) We agree that this is an important point. Towards this, we now demonstrate in Fig. 6A that knockdown of STAU1 results in the stabilization of the *SPATS2* mRNA. Importantly, the destabilization of *SPATS2* mRNA resulting from *SNHG5* depletion can be rescued by removal of STAU1 (Fig. 6B). As expected, the overexpression of *SNHG5* also mediates *SPATS2* stabilization.
- (iv) We further show in Fig. S6E that *SNHG5* can bind directly to the *GLE1* 3'UTR and that this binding can be competed by overexpression of another *SNHG5* target, *SPATS2*.

In the remainder, there are additional comments for the authors to improve the current manuscript for revision.

1. There is a serious disconnect between the two pools of identified *SNHG5* targets depending on which siRNA had been used, as there's only a weak overlap. Does the weak overlap suggest off-target effects?
 - Author response: Indeed, we expect that the limited overlap is suggestive of siRNA off-target effects. We often see this, which is why we concentrate on the overlapping genes only and perform all experiments with at least 2 independent siRNAs. Importantly, we also demonstrate that *SNHG5* overexpressed results in the opposite phenotype from si*SNHG5*,

What about the targets identified by RIA-seq (Fig. 3C). What is the overlap between transcripts deregulated upon *SNHG5* downregulation and RIA-seq results?

- Author response: The overlap between significantly deregulated genes following *SNHG5* inhibition in the RNA-seq and the interacting transcripts identified in the RIA-seq is insignificant. Although we could have expected a significant overlap, the results could reflect that the interacting mRNA overall are lowly expressed and that the majority of the deregulated transcripts from the RNA-seq experiments represent secondary effects and not primary *SNHG5* targets.

And finally, why have the authors selected those 3 top candidates although they started with 121 transcripts? What were the selection criteria?

- Author response: We ranked a short list of *SNHG5*-interacting mRNAs based on the number of peaks (overlapping and not-overlapping) in each transcript and their local enrichment relative to the luciferase background pool (Figure S5D). The mRNA transcripts representing the

peaks were subsequently validated in HCT116 by RIA-qRT-PCR using peak-specific primers (Figure 3d and S5d,e). As evident from Fig. 3D, we identified the *SPATS2*, *PITRM1* and *GLE1* mRNAs as the most enriched *SNHG5* interacting transcripts and hence continued the mechanistic studies focusing on these transcripts.

2. Have the authors investigated whether the two siRNA oligos exert an additive effect when added together? This would be important to know in the context of several figures in the main manuscript.
 - Author response: No we have not analyzed the effect of combining the 2 siRNAs against *SNHG5*. For us the key point has been to have two independent siRNAs to ensure that we were not dealing with off-target effects. To further address the issue of off-target effects, we find it important to perform also the opposite experiment, and indeed overexpression of *SNHG5* mediates the opposite phenotype to the siRNAs.
3. In general, the authors need to provide more experimental detail for their experiments, explain even and odd antisense oligos, be more consistent with figure codes in figures (Fig. 3A versus Supp. Fig. 5a), be more specific in figure legends and methods on describing experiments.
 - Author response: Thank you for pointing this out. We have extended the Materials and Methods section with this in mind, added a Table listing all the primers used and edited the figures and legends for clarity.
4. I would favour moving the GO term analysis from supplements to Fig. 1.
 - Author response: This has been done.
5. Fig. 2a: Labelling is wrong (symbols for siControl, and siSNG52, triangle and circle), the figure legend 2bc reference for transfection to 2a and not 2b.
 - Author response: Thank you. This error has been corrected.
6. In Fig. 2b, the authors reference by mistake "cells were transfected as in b" instead of "as in a"
 - Author response: Thank you. This error has been corrected.
7. The authors should consider changing the order of panels in Fig. 4, as actually in the text, they use the order Fig. 4dc instead of Fig. 4cd.
 - Author response: Thank you for pointing this out. In the revised manuscript the figures now appear in the same order as they are mentioned in the text.
8. In several occasions, key data need to be quantified, e.g. Fig. 4A: Fig. 5, panels A and D and then compare whether the effects are indeed comparable and consistent. Possibly, the effect on the protein level might be stronger than on the RNA level (Fig. 5A).
 - Author response: We agree and western blot quantifications have now

been added to the figure panels 3E, 4A, 4B, 5E, 6A, and 6E.

9. Fig. 3b: there is a problem of scaling, as the right side is lacking any sign of GAPDH bars.

➤ Author response: Indeed, the signals from *GAPDH* are so low that they are not visible with this scale.

10. Fig. 4e: are the authors claiming that there is a correlation between SPATS2 vs SNHG5 as the r value is 0.373? This is not convincing. I would suggest omitting this panel or move into supplements.

➤ Author response: This panel has been moved to supplementary figure S7C.

11. P. 11: Fig. 4 is actually Fig. 4 g.

➤ Author response: Thank you. This error has been corrected.

Reviewer #2 (Remarks to the Author): Expert in CRC

In this manuscript Damas and co-authors describe thier work on lncRNAs in CRC an dspecifically the SNHG5 which is highly upregulated in colon cancer lines. They find that its expression is crucial for cell cycle progression and protection from apoptosis and knockdown induce cell daetah and decreased in vitro and in vivo growth. Using RNAseq they identified 121 associating mRNAs. Further validation revealed that SPATS2 is a target and its mRNA is stabilized by interaction with SNHG5. Indeed KD of SNHG5 results in depletion of SPATS2 in a STAU1 dependent fashion.

Although these data are of interest several points limit enthusiasm and require additional experimentation.

1. It is completely unclear how regulation of SPATS2, a protein with poorly studied characteristics, would drive apoptosis and or cell cycle arrest.
➤ Author response: Indeed, SPATS2 is poorly characterized and has not previously been linked to cancer. Our focus here is on *SNHG5* so a thorough characterization of the mechanisms of SPATS2 in CRC we believe is beyond this manuscript. However, we do agree with the reviewer that additional studies on SPATS2 are warranted in this context to understand the importance of SPATS2 as a key target of *SNHG5* in CRC. In Fig. 5A we now provide evidence that *SPATS2* indeed is up-regulated in CRC compared to paired normal tissue samples. Similar data is provided for *PITRM1* and *GLE1* in Fig. S7a Furthermore, in Fig. 5B, we show that depletion of SPATS2 induces apoptosis in 3 CRC cell lines and thus phenocopies the effect of down-regulating *SNHG5*. Extending on this, we show in Fig. 5C that overexpression of SPATS2 protects CRC cells from oxaliplatin-induced apoptosis, again phenocopying the effects of overexpressing *SNHG5*.

Furthermore, overexpression of SPATS2 partially rescues the effect seen on apoptosis and cell cycle progression following *SNHG5* depletion (Fig. 5E-G). Collectively, our data firmly places SPATS2 downstream *SNHG5* and points to SPATS2 as a protein with important functions in CRC. Additional studies are clearly needed to decipher the mechanism of SPATS2.

In the first figure BCL2/XL are denoted as targets of *SNHG5*. The same holds for STAT3. What is their role in the process? Are these regulated by siSPATS2? The authors should provide more insight in the steps downstream of SPATS2.

- Author response: The western blot panel demonstrating down-regulation of STAT3 and BCL-XL are included in the manuscript to validate the Gene Set Enrichment Analysis performed on the RNA-seq data following *SNHG5* depletion. We do not claim STAT3 or BCL-XL to be direct targets of *SNHG5*. To avoid misunderstandings we have now moved this panel from the main figures to Fig. S3F. We have not analyzed if STAT3 and BCL-XL are affected by SPATS2.
- 2. to claim that SPATS2 is a crucial target requires more extensive proof. Is overexpressing SPATS2 protecting from si*SNHG5* KD? This is needed to make this point.
 - Author response: We agree and this rescue experiment is now presented in Fig. 5E-G.
- 3. Also the effects of si*SNHG5* on SPATS2 in CACO-2 seem rather limited both on cell cycle and cell death. The authors should explain this more carefully and study KD in other CRC lines to confirm their claims. Is it potentially associated with MSI?
 - Author response: To address this point we repeated many of the experiments including an additional CRC cell line, DLD1. This new data is presented in the figures 2A, 2B, 2C, 4A, 5B, S3G, S6A, S6B and S6F. We find that the key phenotypes of manipulating *SNHG5* replicates well across these 3 CRC cell lines. Where differences are found, as for instance in the RIA-seq validation of *PITRM1* and *GLE1*, we believe this to reflect differences in expression levels of these targets in the different CRC cell lines.

Regarding the MSI, we have this data for a limited patient cohort only and we do not see differences in *SNHG5* expression levels. Please see graph below. We have chosen not to include this data in the manuscript. As for the cell lines used, HCT116 and DLD1 are MSI-high whereas CACO2 and HT29 are MSS.

4. KD should also be quantified in the different figures.
 - Author response: Knockdown efficiencies were evaluated in all experiments. Representative examples are shown in the figures 1F (for *SNHG5*), 3E (for *SNHG5*, *PITRM1*, *GLE1* and *SPATS2*), 6A,D (for *STAU1*), S3B (for *SNHG5*), S3G (for *SNHG5*), S4D (for *SNHG5* shRNA) and S6F (for *SPATS2*).

Reviewer #3 (Remarks to the Author): Expert in lncRNAs

The Lund laboratory has identified *SNHG5* as a cytoplasmic long noncoding RNA expressed more abundantly in colorectal cancer cells. The authors show that *SNHG5* associates with a number of mRNAs in cells, including *SPATS2*, *PITRM1* and *GLE1* mRNAs, and propose that it renders them stable by competing for the binding of the decay-promoting RNA binding protein Staufen (*STAU*). The topic of this manuscript is interesting, timely, and novel. However, some of the data need to be strengthened, particularly the data

pertaining to the involvement of STAU.

Main comments

1. Introduction: The authors state that the only cytoplasmic lncRNAs described are ceRNAs, but in fact this characterization is incomplete. They need to mention other examples of cytoplasmic lncRNAs, including other examples of lncRNAs associating with mRNAs described in the literature.
 - Author response: We do not claim that ceRNAs are the only described cytoplasmic lncRNAs. We state (on page 4) that much less is known about the functions of cytoplasmic lncRNAs and that a noticeable exception from this void of knowledge is ceRNAs.
2. How many copies of SNHG5 are there per cell? How does this number compare with the numbers of SPATS2, PITRM1 and GLE1 mRNAs?
 - Author response: We agree that molecule number is an important parameter - especially for lncRNAs, which often are found lowly expressed. Furthermore, for *SNHG5* to act in a competitive fashion with STAU1 on its targets, we could expect *SNHG5* and the targets to be expressed at somewhat similar levels. To address this we *in vitro* transcribed *SNHG5* and *SPATS2* RNAs and used this to generate a standard curve where a known number of molecules were amplified using qRT-PCR. qRT-PCR data from the CRC cell lines were subsequently compared to the standard curve. This data is presented in Fig. S6C and show that *SNHG5* and *SPATS2* indeed are expressed at similar levels in HCT116, DLD-1 and CACO2 CRC cells.
3. The interaction between SNHG5 and SPATS2, PITRM1 and GLE1 mRNAs needs to be tested in vitro. What are the regions of complementarity that mediate the interaction of SNHG5 with PITRM1 and GLE1 mRNAs? These should be identified as well. The SNHG5 mutant lacking the region of interaction with SPATS2 mRNA should be tested in vitro and in vivo. Does the mutant affect tumorigenesis?
 - Author response: We agree that this is an important point. Towards this we generated reporter vectors holding fragments from the 3'UTRs of *SPATS2* and *GLE1* and tested the effect on the reporters of overexpressing *SNHG5* or a mutant of *SNHG5* lacking exon 1, which encompasses the interaction region. This data is presented in Fig. 4E and S6E. We also included a competition experiment validating that the effect of *SNHG5* on the *GLE1* reporter can be competed by adding another target, *SPATS2* (Fig. S6E). The interaction sites of *SNHG5* on *SPATS2* and *GLE1* are presented in Fig. 4C and Fig. S6. To identify the region of *SNHG5* responsible for the biological effect we constructed a series of deletion mutants (Fig. 2F,G). These mutants were subsequently used to query the effect on target (*SPATS2*) mRNA

stability (Fig. 4D), which pointed us to exon 1 of *SNHG5*. As our *in vivo* model employs *SNHG5* knockdown, we did not evaluate if the exon 1 mutant affect tumorigenesis *in vivo*.

4. The data in Figure 5 need to be developed further:

- the authors do not explain why they decided to focus on STAU and not on many other RNA binding proteins that can promote mRNA decay. Do other such proteins affect *SNHG5* function? In fact, they could also have pursued microRNAs to explain this effect. They should explain why STAU was chosen.

➤ Author response: We focused our attention on STAU1 since it has previously been shown to destabilize mRNAs in the cytoplasm (Park and Maquat, 2013). Also, several of the mRNA targets for *SNHG5* identified in the RIA-seq were previously found to interact with STAU1 (Sugimoto et al. 2015). This lead us to perform the rescue experiment shown in Fig. 6E.

- the authors show that STAU binds *SPATS2* mRNA, but because the IgG values are not adjusted, it is difficult to see the consequences of STAU binding to *SPATS2* mRNA after *SNHG5* silencing. The levels of mRNAs in IgG IP (after normalization to a nonspecific RNA) should be adjusted to 1, so that the enrichments in STAU IP samples can be compared.

➤ Author response: The levels of mRNAs in the IgG IP have now been adjusted to 1 (Fig. 6C).

Does the interaction of STAU with *SPATS2* mRNA increase when *SNHG5* is overexpressed? Does it remain unchanged when the *SNHG5* deletion mutant is overexpressed?

➤ Author response: We would expect the interaction of STAU1 with *SPATS2* mRNA to *decrease* with *SNHG5* overexpression. This exact experiment was for technical reasons not feasible to perform. The experiments presented in Fig. 6C are based on cells from 10 15 cm plates or 4 625cm² plates for each sample. It would not be possible to conduct transient *SNHG5* (and mutant) overexpression at this scale. However, we queried the STAU1/*SPATS2*/*SNHG5* inter-dependencies by several other means:

- a. In Fig. 3, Fig 4. And Fig. S6 we demonstrate that knockdown and overexpression of *SNHG5* affects both the mRNA and protein levels of *SPATS2*.
- b. In the revised Fig. 5, we demonstrated that both knockdown and overexpression of *SPATS2* phenocopies the effects seen when manipulating the levels of *SNHG5*.
- c. In the new Fig. 6 we show that loss of STAU1 leads to an increase in *SPATS2* mRNA levels and that the reduction in *SPATS2* mRNA levels seen after *SNHG5* depletion can be rescued by removal of STAU1.
- d. In Fig. 6C we demonstrate that the amount of *SPATS2* mRNA bound to STAU1 is increased following *SNHG5* depletion.

- What region of SPATS2 mRNA does STAU bind to? Is the site of interaction of SPATS2 mRNA with the SNHG5 the same as the site of interaction with STAU?
 - Author response: To answer this question we would have to perform a STAU1 CLIP experiment or mobility-shift assays using a panel of *SPATS2* mutants and super-shifting with anti-STAU1. As our focus here is on *SNHG5*, we have not pursued these experiments.

- Does STAU also bind PITRM1 and GLE1 mRNAs? Are these interactions antagonized in a similar manner by SNHG5?
 - Author response: STAU1 has previously been published to bind *PITRM1* and *GLE1* in HEK293 cells (Sugimoto et al. 2015), however, we have not been able to demonstrate this interaction reproducibly, likely because the expression levels of *PITRM1* and *GLE1* are significantly lower than that of *SPATS2* in the CRC cells employed here.

Minor comments

Fig S2e: In the Y axis, please fix a typo (it is 'Retrieved' not 'Retreived')

- Author response: This has been corrected.

Please use HCT116 consistently (HCT-116 is less commonly used)

The authors need to adopt the appropriate nomenclature for human and mouse genes, lncRNAs, mRNAs, and proteins.

- Author response: This has been corrected.

Reviewer #1 (Remarks to the Author)

As outlined in my initial review, this potentially interesting study identifies SNHG5 as a cytoplasmic, long non-coding RNA with increased expression in colorectal cancer. When knocked out, apoptosis is induced while overexpression protects cancer cells from oxaliplatin-induced apoptosis. The authors identified amongst other target RNAs SPATS2, which also causes – when deleted – apoptosis. In the revised version, the authors now strengthen the part reg. SNHG5 and SPATS2, as overexpression of SPATS2 partially rescues the apoptotic phenotype resulting from SNHG5 depletion (Fig. 5). Furthermore, they now provide data suggesting that Stau1 might be functionally involved in this process as down-regulation of Stau1 reverses the effect of SNHG5 on SPATS2 mRNA levels (Fig. 6C).

It is unfortunate that the authors decided not to choose one of the two options for a revised version of their study. Whereas the part on SNHG5 and SPATS2 is strong and deserves high priority for publication, the weak and still non-convincing part of the study is the one on Stau1 and its mechanistic role in the process. Although the authors could now demonstrate that Stau1 is involved, it is not clear at all whether this is due to Stau1 mediating SPATS2 mRNA destabilization. Consequently, the title of the study (and also the abstract) must be changed as the data presented in the revised version do not support this notion: “SNHG5 ... counteracts STAU1-mediated mRNA stabilization.” Also, there is actually at least one alternative and more likely mechanism of Stau1 action. Their new Fig. 6B shows an effect of approx. 15% at the mRNA target level (Stau1 downregulation). However, there might be a significantly higher effect on the SPATS2 protein level (shown in new Fig. 6A). This is actually more likely to be the case. Here, I strongly recommend to also include the following new citations, e.g. recent published work by Jernej Ule in Nature (Sugimoto et al., 2015) and to Melissa Moore in Nature SMB (Ricci et al., 2014: this paper corrects previous findings by another lab that Stau1 actually destabilizes transcripts!).

Importantly, the authors should provide experimental insight into the fact whether Stau1 indeed binds SPATS2, e.g. whether SPATS2 indeed has hiCLIP sites for Stau1.

Finally, there is a key mis-annotation of the y-axis in the new Fig. 6C. This is very likely NOT enrichment relative to INPUT, but to IgG.

Reviewer #2 (Remarks to the Author)

The revisions of the manuscript have clearly improved the manuscript. Unfortunately the authors have chosen to take out the observations on BCL2/XL and STAT3 instead of improving on these observations. This rather weakened the manuscript in the sense that it has become increasingly unclear how STAU1 regulates the cell death observed. Although the data on SNHG5/SPATS2 are clear the manuscript still fails to provide insight on how this regulates the changes in cell cycle and cell death. In this reviewer's view that is needed to increase the novelty and mechanistic insight.

This could be achieved by simply analyzing whether the regulation of BCL2/XL are needed and also observed in SPATS2 KD cells

Reviewer #3 (Remarks to the Author)

The authors' responses are satisfactory.

Rebuttal letter to the 2nd manuscript revision.

Please consult also the rebuttal letter accompanying the 1st manuscript revision for a complete overview of the amount of new data added.

Reviewers' comments:

Reviewer #1 (Remarks to the Author):

As outlined in my initial review, this potentially interesting study identifies SNHG5 as a cytoplasmic, long non-coding RNA with increased expression in colorectal cancer. When knocked out, apoptosis is induced while overexpression protects cancer cells from oxaliplatin-induced apoptosis. The authors identified amongst other target RNAs SPATS2, which also causes – when deleted – apoptosis. In the revised version, the authors now strengthen the part reg. SNHG5 and SPATS2, as overexpression of SPATS2 partially rescues the apoptotic phenotype resulting from SNHG5 depletion (Fig. 5). Furthermore, they now provide data suggesting that Stau1 might be functionally involved in this process as down-regulation of Stau1 reverses the effect of SNHG5 on SPATS2 mRNA levels (Fig. 6C).

It is unfortunate that the authors decided not to choose one of the two options for a revised version of their study.

Author reply: This reviewer originally suggested to either focus on the clinical aspects and remove the original figure 5 dealing with the mechanism involving STAU1, or extend the mechanistic analysis. As outlined in the rebuttal letter to the 1st manuscript revision we did both:

A) We added more clinical data on the expression of SPATS2, PITRM1 and GLE1 in CRC. This data is presented in Fig. 5a and Fig S7. In accordance with our hypothesis, all 3 SNHG5 targets are indeed up-regulated in CRC.

B) We substantially extended the mechanistic analyses of both STAU1 and SPATS2 as outlined in the manuscript and the rebuttal letter.

Whereas the part on SNHG5 and SPATS2 is strong and deserves high priority for publication, the weak and still non-convincing part of the study is the one on Stau1 and its mechanistic role in the process. Although the authors could now demonstrate that Stau1 is involved, it is not clear at all whether this is due to Stau1 mediating SPATS2 mRNA destabilization. Consequently, the title of the study (and also the abstract) must be changed as the data presented in the revised version do not support this notion: “SNHG5 ... counteracts STAU1-mediated mRNA stabilization.”

Author reply: We disagree with the reviewer. Our data show:

A) As evident from Fig 6C, STAU1 binds directly to SPATS2 and loss of SNHG5 diminishes this interaction. This evidence was originally disputed by Reviewer 1, likely due to a misunderstanding regarding the crosslinking procedure, but this is fortunately no longer so.

B) As evident from Fig 6A, knockdown of STAU1 results in a 60% increase in the SPATS2 mRNA level. This is also reflected at the SPATS2 protein level (Fig 6A,D) and in the functional studies presented in the Fig 6E.

C) As evident from Fig 6B, loss of STAU1 rescues the increased turnover of SPATS2 mRNA following SNHG5 depletion.

Hence, our data demonstrate both direct binding and regulation. We therefore see no reason to change the title or abstract, but we are willing to do so if you find it important.

Also, there is actually at least one alternative and more likely mechanism of Stau1 action. Their new Fig. 6B shows an effect of approx. 15% at the mRNA target level (Stau1 downregulation). However, there might be a significantly higher effect on the SPATS2 protein level (shown in new Fig. 6A). This is actually more likely to be the case.

Author reply: We believe the reviewer is misreading the graphs. Fig 6A (right panel) demonstrates a 60% increase in SPATS2 mRNA levels following STAU1 depletion (not 15% as stated by the reviewer), and an 80% increase in SPATS2 protein level (left panel). Although the difference between 60% and 80% could reflect an effect of STAU1 on SPATS2 translation, the difference is so small it could just reflect the variation observed between two different methodologies (qRT-PCR and western blotting). The Fig 6B referred to by the reviewer shows the changes in SPATS2 mRNA turnover rate (which is approximately 15% at this specific time-point) – not the effect on steady state SPATS2 mRNA levels.

To accommodate this reviewer concern, we have inserted a sentence in the discussion stating that we cannot rule out that STAU1 may also affect SPATS2 translation.

Here, I strongly recommend to also include the following new citations, e.g. recent published work by Jernej Ule in Nature (Sugimoto et al., 2015) and to Melissa Moore in Nature SMB (Ricci et al., 2014: this paper corrects previous findings by another lab that Stau1 actually destabilizes transcripts!).

Author reply: We actually already refer to both of these studies. They are listed as references 15 and 17, respectively. We are aware that different roles for STAU1 have been described in different systems. We, however, have to report in accordance with the data we obtain from CRC cells.

Importantly, the authors should provide experimental insight into the fact whether Stau1 indeed binds SPATS2, e.g. whether SPATS2 indeed has hiCLIP sites for Stau1.

Author reply: This experimental evidence is presented in Fig 6C and backed up by functional studies in Fig 6A, B, D, E. Regarding the question of hiCLIP sites for STAU1 on SPATS2, we analyzed the publically available data (Sugimoto et al., 2015) and did not find such sites. This, however, is unsurprising as that study was performed in HEK293 cells in which SPATS2 is very lowly expressed. Please notice that producing a STAU1 hiCLIP dataset on its own was a Nature paper last year.

Finally, there is a key mis-annotation of the y-axis in the new Fig. 6C. This is very likely NOT enrichment relative to INPUT, but to IgG.

Author reply: That is correct. In the first round of revision Reviewer 3 requested a re-plot of this panel, which we performed. Unfortunately, we did not correct the Y axis title accordingly. This has now been corrected.

Reviewer #2 (Remarks to the Author):

The revisions of the manuscript have clearly improved the manuscript. Unfortunately the authors have chosen to take out the observations on BCL2/XL and STAT3 instead of improving on these observations. This rather weakened the manuscript in the sense that it has become increasingly unclear how STAU1 regulates the cell death observed. Although the data on SNHG5/SPATS2 are clear the manuscript still fails to provide insight on how this regulates the changes in cell cycle and cell death. In this reviewer's view that is needed to increase the novelty and mechanistic insight.

This could be achieved by simply analyzing whether the regulation of BCL2/XL are needed and also observed in SPATS2 KD cells

Author reply: The observations on STAT3/BCL2/BCL-XL have not been taken out – merely moved to Fig S3F. As stated in the rebuttal letter, these blots are included to validate the Gene Set Enrichment Analyses presented in Fig S3E.

However, we agree with the reviewer that analyzing the effects of silencing SPATS2 on STAT3 and BCL-XL would strengthen the manuscript. Accordingly, we now demonstrate in a new Fig. S7c, that the expression of both STAT3 and BCL-XL is markedly lower in cells depleted of SPATS2. Hence, as this phenocopies the effects seen following silencing of SNHG5, the data further underscores SPATS2 as a key downstream target of SNHG5 in CRC.

Reviewer #3 (Remarks to the Author):

The authors' responses are satisfactory.

Reviewer #1 (Remarks to the Author)

The authors made a serious attempt to address the concerns of the reviewers. Consequently, the revised manuscript considerably improved. In sum, I found the responses by the authors to be satisfactory.

By the way, there was a misunderstanding by the authors. By no means, it was intended to ask for a new set of hiCLIP data. The suggestion has simply been to check out the results by Sugimoto and Ule, 2015, Nature. I still favor the idea of including the following statement from the authors:

„Regarding the question of hiCLIP sites for STAU1 on SPATS2, we analyzed the publically available data (Sugimoto et al., 2015) and did not find such sites. This, however, is unsurprising as that study was performed in HEK293 cells in which SPATS2 is very lowly expressed.

Minor point:

- In the text, the authors wrote: „ cells in Fig. 6b were treated for 5 hours with triptolide. However, in the material and methods section, they say 8 hours.

Rebuttal letter

Reviewer #1 (Remarks to the Author):

The authors made a serious attempt to address the concerns of the reviewers. Consequently, the revised manuscript considerably improved. In sum, I found the responses by the authors to be satisfactory.

By the way, there was a misunderstanding by the authors. By no means, it was intended to ask for a new set of hiCLIP data. The suggestion has simply been to check out the results by Sugimoto and Ule, 2015, Nature. I still favor the idea of including the following statement from the authors:

„Regarding the question of hiCLIP sites for STAU1 on SPATS2, we analyzed the publically available data (Sugimoto et al., 2015) and did not find such sites. This, however, is unsurprising as that study was performed in HEK293 cells in which SPATS2 is very lowly expressed.

Author reply: This sentence has been inserted into the Discussion

Minor point:

- In the text, the authors wrote: „ cells in Fig. 6b were treated for 5 hours with triptolide. However, in the material and methods section, they say 8 hours.

Author reply: This has been corrected